# PREFERENCE DISCERNING IN GENERATIVE SEQUENTIAL RECOMMENDATION

## ABSTRACT

Sequential recommendation systems aim to provide personalized recommendations for users based on their interaction history. To achieve this, they often incorporate auxiliary information, such as textual descriptions of items and auxiliary tasks, like predicting user preferences and intent. Despite numerous efforts to enhance these models, they still suffer from limited personalization. To address this issue, we propose a new paradigm, which we term *preference discerning*. In *preference discerning*, we explicitly condition a generative sequential recommendation system on user preferences within its context. The user preferences are generated by Large Language Models (LLMs) based on user reviews. To evaluate preference discerning capabilities of sequential recommendation systems, we introduce a novel benchmark that provides a holistic evaluation across various scenarios, including preference steering and sentiment following. We assess current state-of-the-art methods using our benchmark and show that they struggle to accurately discern user preferences. Therefore, we propose a new method named Mender (**M**ultimodal Prefer**en**ce **D**iscern**er**), which improves upon existing methods and achieves state-of-the-art performance on our benchmark. Our results show that Mender can be effectively guided by human preferences, paving the way toward more personalized sequential recommendation systems. We will open-source the code and benchmarks upon publication.

## 1 INTRODUCTION

Sequential recommendation is the task of recommending items to a user based on their historical interactions. This requires inferring latent variables, such as user preferences and intent, which are often not explicitly provided in publicly available datasets. To improve personalization, several sequential recommendation systems leverage auxiliary information, including heterogeneous interaction types (Meng et al., 2020), item descriptors (e.g., textual, visual) (Hidasi et al., 2016b; Liu et al., 2021; Zhang et al., 2019a), temporal information (Bogina & Kuflik, 2017; Li et al., 2020), and keyword-based user queries (He et al., 2022). By incorporating such information, these systems can better approximate a user's intent, leading to improved performance and personalization.

Although incorporating additional information can provide benefits, the degree of personalization in current sequential recommendation models remains limited. User decisions on what item to acquire next are guided by their preferences, which are typically not explicitly provided in commonly used recommendation datasets (Ni et al., 2019; Zhang et al., 2015). As a result, these preferences must be approximated from the user's interaction history. Recent works have leveraged LLMs to extract user preferences from existing datasets and leverage them for auxiliary tasks (Zhang et al., 2023; Cao et al., 2024). However, these approaches do not allow the model to be dynamically steered by user preferences in their context during inference. Furthermore, there is currently no benchmark that effectively evaluates to what extent those models discern preferences.

To address these limitations, we propose a novel paradigm, which we term *preference discerning*. preference discerning entails training a multimodal generative retrieval model conditioned on user preferences within its context (see Figure 1). This requires approximating a user's preference in textual form from user-specific data, such as reviews via pre-trained LLMs (Kim et al., 2024). By conditioning the sequential recommendation system on user preferences in-context, we unlock steering via generated user preferences, effectively combining the sequential prior from interaction history

Figure 1: The preference discerning paradigm, comprising two components: *preference approximation* and *preference conditioning*. Preference approximation utilizes pre-trained LLMs to infer user preferences from user-specific data. In preference conditioning a sequential recommendation system is conditioned on the generated user preferences, enabling personalized recommendations.

with the user preferences. Therefore, users can specify in natural language what item properties they wish to avoid or prefer. The sequential recommendation system then integrates this information with past interactions to provide a well-personalized recommendation.

To evaluate preference discerning capabilities of sequential recommendation systems, we propose a holistic benchmark that comprises five evaluation axes: (1) preference-based recommendation, (2) sentiment following, (3) fine-grained steering, (4) coarse-grained steering, and (5) history consolidation. We evaluate state-of-the-art generative retrieval methods on our benchmark and find they lack several key abilities of preference discerning. Therefore, we introduce a novel multimodal generative retrieval method named **M**ultimodal prefer**en**ce **d**iscern**er** (Mender) which effectively fuses pre-trained language encoders with the generative retrieval framework for preference discerning. We demonstrate that preference discerning capabilities can naturally emerge when training solely on preference-based recommendation data. Furthermore, we show that preference discerning capabilities can be obtained by augmenting the training data with training splits for the different axes. As a result, Mender can be effectively steered by different user preferences provided in it's context to recommend specific items. Ultimately, Mender mostly outperforms existing state-of-the-art generative retrieval models across all evaluation axes of our benchmark. In summary, our contributions are as follows:

- We introduce a novel paradigm called *preference discerning*, where the generative sequential recommendation system is conditioned on user preferences within its context.

- We propose a comprehensive benchmark for evaluating preference discerning, comprising of five distinct evaluation scenarios that provide a holistic assessment of its capabilities.

- We present Mender, a multimodal baseline that integrates collaborative semantics with language preferences, achieving state-of-the-art performance on our proposed benchmark.

## 2 RELATED WORK

**Sequential Recommendation** can be categorized into two major scenarios: search (Nigam et al., 2019) and recommendation (Covington et al., 2016). The former assumes access to a query from a user that reflects their intent (He et al., 2022), whereas the latter scenario does not make such an assumption. For the recommendation scenario, numerous works have investigated the use of additional information to enhance recommendation performance (Meng et al., 2020; Hidasi et al., 2016a; Liu et al., 2021; Zhang et al., 2019a; Bogina & Kuflik, 2017; Li et al., 2020). Our work introduces a new paradigm that enables in-context steering of sequential recommendation systems by textual user preferences.

**Existing Benchmarks** for recommendation vary in their representation of user preferences and the tasks they evaluate. Oh et al. (2024) proposed a benchmark for instruction-following in information retrieval where instructions are generated from user-specific data via LLMs. The C4 benchmark (Hou et al., 2024) uses complex search queries that reflect user preferences for retrieval. Contrary, we focus on user preferences in sequential recommendation. Such preferences are often modeled indirectly from user queries and responses to recommended items (Min et al., 2023; Huang et al., 2013; Ma et al., 2018), or represented as edges on graphs (Ying et al., 2018; Li et al., 2019). In query-aware sequential recommendation He et al. (2022) the model is given keywords in its context

that represent the user's intent but do not capture their preferences. In contrast, our benchmark builds on established datasets (Ni et al., 2019; Kang & McAuley, 2018) and augments them with generated user preferences to evaluate preference discerning capabilities.

**Generative Retrieval** uses autoregressive modeling to generate the next item, rather than performing pairwise comparisons between a user representation and all item representations. Rajput et al. (2023) proposes tokenizing items in the form of semantic IDs (Lee et al., 2022). The benefit of this approach is that very large item sets can be represented as a combination of ids that reflect their semantic similarity. Subsequent works have investigated the effect of learned tokenization (Sun et al., 2023) and additional objectives (Li et al., 2024; Wang et al., 2024). Our Mender represents items as semantic IDs and fuses them with pre-trained LMs to effectively steer the recommendation.

**Language-Based Sequential Recommendation** rely on the premise of enhanced transparency and actionable interrogation of recommendation systems (Radlinski et al., 2022). Furthermore, language provides a natural interface for users to express their preferences and allows harnessing the expressive power of LLMs. Recent works have leveraged LLMs to approximate user preferences by representing user-and item-specific data in natural language (Zheng et al., 2023; Oh et al., 2024; Sanner et al., 2023; Cao et al., 2024),by conditioning the LLM on user embeddings (Ning et al., 2024), or by leveraging user reviews for items (Kim et al., 2024). In this context, Kang et al. (2023) found that effective preference approximation may require fine-tuning of the LLM. Other studies have explored using LLMs for data augmentation in sequential recommendation (Geng et al., 2022; Zhang et al., 2019b; Luo et al., 2024). In the near cold-start scenario, Sanner et al. (2023) demonstrated that user preferences represented in natural language can be particularly effective. Li et al. (2023) showed the benefit of moving from ID-based representations to text-based representation of the interaction history. Similarly, Petrov & Macdonald (2023) represent all items in natural language and performs ranking conditioned on past interactions. Zheng et al. (2023) explored aligning semantic IDs with natural language by adding auxiliary tasks. The key difference to our Mender is that it operates on different levels of abstraction to represent the user's interaction history.

## 3 METHODOLOGY

The *preference discerning* paradigm comprises two primary components: *preference approximation* and *preference conditioning* (see Figure 1).

### 3.1 PREFERENCE APPROXIMATION

Preference approximation refers to the process of inferring a user's preferences based on user- and item-specific data. This process has been User-specific data may include user reviews, profiles, posts, demographic information, or any other relevant details. The incorporation of item-specific information is crucial, as it provides additional context that can help alleviate the vagueness or incompleteness often encountered in user-specific data. Preference approximation is a necessary pre-requisite that enables in-context conditioning on the generated user preferences.

In the context of sequential recommendation, we assume access to a set of users $\mathcal{U}$ and a set of items $\mathcal{I}$. For each user $u \in \mathcal{U}$, we are given a sequence of item purchases in chronological order: $s_u = \left[i_u^1, \ldots, i_u^{T_u}\right]$, where $T_u$ represents the time horizon for a particular user $u$ who has purchased items $i \in \mathcal{I}$. The task of the sequential recommendation system is to predict the next item based on the interaction history. We also assume

---

**Algorithm 1** Preference Approximation

**Input:** prompt $x$, users $\mathcal{U}$, items $\mathcal{I}$, reviews $\mathcal{R}$, Language Model LLM$(\cdot)$
1: **for** $u \in \mathcal{U}$ **do**
2:     **for** $t \in \{1, \ldots, T_u\}$ **do**
3:         $\mathcal{P}_u^{(t)} \leftarrow \text{LLM}\left(\left[x; i_u^{(1)}; r_u^{(1)}; \ldots; i_u^{(t)}; r_u^{(t)}\right]\right)$
4:     **end for**
5: **end for**

---

access to user-specific data including user reviews $r \in \mathcal{R}$ and natural language descriptions of items. For each user $u$ and for each timestep $1 \leq t \leq T_u$, we collect reviews $\{r_u^{(1)}, \ldots, r_u^{(t)}\}$ along with item information $\{i_u^1, \ldots, i_u^{(t)}\}$ from their interaction history $s_u$ and prompt an LLM to approximate the user's preferences. We additionally add a prompt $x$ to the interaction history which contains gen-

eral instructions such as to neglect aspects like delivery time or pricing and encode aversions of the user. With this process we obtain a set of five user preferences $\mathcal{P}_u^{(t)}$ for each each timestep $t$ that is based on the past interaction history. Importantly, the information contained in the different user preferences are mostly orthogonal, i.e. they describe preferences with respect to different items, or item properties (see an example in Appendix C). To verify the quality of the generated preferences we conduct a user study (see Appendix F). The participans found that usually around 75% of the generated preferences correctly approximate the user's preferences. A schematic illustration of the preference generation procedure is shown in Figure 9 and we provide pseudocode in Algorithm 1. For details on prompts, generation process, or preference granularity, we refer to Appendix C.

## 3.2 BENCHMARK GENERATION

We compile a comprehensive benchmark that enables a holistic evaluation of preference discerning capabilities. To achieve this, we define five axes of evaluation: *Preference-based recommendation*, *Sentiment following*, *Fine-grained steering*, *Coarse-grained steering*, and *History consolidation*. Below, we elaborate on each of these axes and discuss their respective use cases.

**Preference-based Recommendation.** This evaluation scenario extends the sequential recommendation scenario by incorporating the generated user preferences. For this task, the model receives a single user preference of the set $\mathcal{P}_u^t$ along with the interaction history and must predict the next item $i_t$. We select the preference that yields the maximum cosine similarity to $i_t$ in a pre-trained sentence embedding space (Ni et al., 2022). More formally, given a pre-trained sentence embedding model $\phi(\cdot)$, we select $p_u^{(t-1)}$ as

$$p_u^{(t-1)} = \arg \max_{p \in \mathcal{P}_u^{(t)}} \frac{\phi(p)^\top \phi(i_t)}{\|\phi(p)\|\|\phi(i_t)\|}. \tag{1}$$

This results in a setting where each ground-truth item $i_u^{(t)}$ is associated with a single user preference $p_u^{(t-1)}$. Therefore, the input to the sequential recommendation system is a sequence of $\left[p_u^{(t-1)}, i_u^{(1)}, \ldots, i_u^{(t-1)}\right]$ and the task is to predict $i_u^{(t)}$. Since $p_u^{(t-1)}$ is generated based only on information about past items, there is no information leak that could reveal the ground truth item, i.e. there is no information leak and the underlying aleatoric uncertainty of the task is preserved.

**Fine-Grained & Coarse-Grained Steering.** This axis can be useful in leveraging organic data for, e.g., recommending ads. As an example, if a user is an opponent of exercise and fitness and engages in such discussion, a model can steer the recommendations such that they avoid weight-loss medications even if the user has purchased them in the past. Recall, that the preference-based recommendation scenario captures the underlying uncertainty of the original recommendation task as we provide the model with $p_u^{(t-1)}$ to predict $i_u^{(t)}$. This can result in cases where $p_u^{(t-1)}$ is not semantically related to $i_u^{(t)}$, as oftentimes $i_u^{(t)}$ is not related to previously acquired items. However, our aim is to quantify how well the model can *follow* the user preference to recommend certain items. The intuition is that if a user provides additional information about their preferences to the recommendation system, the system should adapt its recommendation accordingly. Therefore, our goal is to quantify the model's ability to be steered towards items that are either very similar or very distinct from $i_t$ by modifying the user preference in its context. To achieve this, we identify a very similar item $\tilde{i}^{(t)}$ and a very distinct item $\hat{i}_u^{(t)}$ to the ground-truth item $i_u^{(t)}$ by

$$\tilde{i}_u^{(t)} = \arg\max_{i \in \mathcal{I}\setminus\{i_u^{(t)}\}} \frac{\phi(i)^\top \phi(i_u^{(t)})}{\|\phi(i)\|\|\phi(i_u^{(t)})\|}, \quad \text{and} \quad \hat{i}_u^{(t)} = \arg\min_{i \in \mathcal{I}\setminus\{i_u^{(t)}\}} \frac{\phi(i)^\top \phi(i_u^{(t)})}{\|\phi(i)\|\|\phi(i_u^{(t)})\|}. \tag{2}$$

Next, we associate $\tilde{i}^{(t)}$ and $\hat{i}_u^{(t)}$ with different user preferences $p_1$ and $p_2$ by

$$p_1 = \arg\max_{p \in \mathcal{P}} \frac{\phi(p)^\top \phi(\tilde{i}^{(t)})}{\|\phi(p)\|\|\phi(\tilde{i}^{(t)})\|}, \quad \text{and} \quad p_2 = \arg\max_{p \in \mathcal{P}} \frac{\phi(p)^\top \phi(\hat{i}_u^{(t)})}{\|\phi(p)\|\|\phi(\hat{i}_u^{(t)})\|}, \tag{3}$$

where $\mathcal{P}$ denotes the sett of accumulated preferences across all users and items. Additionally, we obtain a target user $\hat{u}$ with the same ground truth item $i_{\hat{u}}^{(t)} = i_u^{(t)}$, but a different interaction history. The motivation for this is to enhance the variability in the generated datasets. By combining these elements, we create two new sequences: $\left[p_1, i_{\hat{u}}^{(1)}, \ldots, i_{\hat{u}}^{(t-1)}\right]$ and $\left[p_2, i_u^{(1)}, \ldots, i_u^{(t-1)}\right]$ with ground

truth items $\tilde{i}_u^{(t)}$, and $\hat{i}_u^{(t)}$, respectively. A visual illustration of this procedure is provided in Figure 12. Throughout the dataset creation process, we ensure that the preferences used during training are not associated with the evaluation items. This allows us to evaluate the model's ability to generalize and respond to new preferences that are semantically similar to preferences in the training set.

**Sentiment Following.** This axis is crucial for utilizing organic data. For example, on social media, we have access to the users' interactions with ads, but also to organic data such as posts, comments, and likes. A user may discuss on their posts or comments that they do not like a specific brand of phone but then they may accidentally click on an ad for the same brand of the phone. Sentiment following allows the system to utilize negatively formulated user preferences to correctly identify what items *not* to retrieve. To evaluate this scenario, we instruct the LLM during preference approximation to generate preferences that contain information about items that should *not* be retrieved. To identify *negative* preferences and reviews, we classify them using pre-trained sentiment classification models. Then, we match negative preferences with items that received negative reviews, as these most likely elicited the negative preference (see Figure 10). The matching is done via cosine similarity in Sentence-T5 space. This results in tuples of $(p_u^-, i)$, where $p_u^-$ represents a negative preference and $i$ is the matched item. To obtain a positive pair $(p_u^+, i)$, we apply a rule-based inversion of the negative preference (Figure 11). The details of this rule-based inversion are provided in Appendix D. Notably, the compiled data consists solely of $(p_u^\pm, i)$ tuples, without interaction history.

To evaluate this scenario we rely on a combined hit-rate measure. Given a set of k predicted candidate items $\mathcal{C} = \{\bar{i}_1, \ldots, \bar{i}_k\}$, we check whether the ground truth item occurs in $\mathcal{C}$, i.e. $\mathbb{1}_\mathcal{C}(i) = 1$, where $\mathbb{1}(\cdot)$ represents the indicator function. Now let us assume that we obtain two prediction sets $\mathcal{C}^+$ and $\mathcal{C}^-$, where $\mathcal{C}^+$ is obtained using the positive preference $p_u^+$ and $\mathcal{C}^-$ by using the negative preference $p_u^-$ for item $i$. Then the combined hit-rate measure can be computed as $m = \mathbb{1}_{\mathcal{C}^+}(i) \wedge \neg \mathbb{1}_{\mathcal{C}^-}(i)$. Here $m = 1$ indicates that the model successfully retrieved the item for $p_u^+$, while simulaneously *not* predicting it for $p_u^-$. This measure can then again be computed for different sizes of prediction sets, i.e. $m@k$, as conventional retrieval metrics.

**History Consolidation.** User preferences may change over time and users usually have different preferences that relate to different items. For example a user may prefer running shoes based on a certain foam but also prefers lightness. Consider that after some time the kind of foam may not be as important to the user anymore. Then, the recommendation system should be capable of adapting its recommendation based on the interaction history and be able to ignore preferences it has received originally. Therefore, we aim to evaluate the capabilitiy of the system to incorporate multiple user preferences and ignore some of them. To simulate such a use case, we leverage he fact that the generated user preferences are mostly orthogonal and provide the whole set of five generated preferences $\mathcal{P}_u^{(t-1)}$ to the model simultaneously where the task is to predict the ground-truth item $i_u^{(t)}$. The preferences in $\mathcal{P}_u^{(t-1)}$ are usually orthogonal in the information they provide (see an example in Appendix C). Therefore, they are not necessarily valuable to make a more accurate prediction. In fact, this evaluation scenario can be considered more difficult than preference-based recommendation, as it incorporates both time dependency, as well as a higher content of noise in the preferences. In this evaluation scenario the originally matched preference is contained in $\mathcal{P}$. Therefore, in order to accurately predict the ground truth item, the model most infer the matched preference out of $\mathcal{P}$. The corresponding evaluation sequences are structured as $\left[ p_{u_1}^{(T_u-1)}, \ldots, p_{u_5}^{(T_u-1)}, i_1, \ldots, i_u^{(T_u-1)} \right]$ and contain all five generated user preferences.

### 3.3 MULTIMODAL PREFERENCE DISCERNER (MENDER)

We propose Mender, a novel multimodal generative sequential recommendation system. Mender can be conditioned on user preferences expressed in natural language in its context and generates item identifiers. Mender builds on the recently proposed TIGER (Rajput et al., 2023), a generative retrieval model trained on semantic IDs. These semantic IDs are obtained by training a RQ-VAE (Lee et al., 2022) on item embeddings in Sentence-T5 space. Given an item embedding $\boldsymbol{e} \in \mathbb{R}^d$, the RQ-VAE quantizes $\boldsymbol{e}$ into a discrete feature map as:

$$\mathcal{RQ}(\boldsymbol{e}, \mathcal{C}, D) = (k_1, \ldots, k_N) \in [K]^N \tag{4}$$

where $\mathcal{C}$ represents a finite set of tuples $\{(k, \boldsymbol{c}_k)\}_{k \in K}$, $K$ denotes the granularity of the codebook $\mathcal{C}$, and $N$ corresponds to the depth of the RQ-VAE, i.e., the number of codebooks. A user sequence

$s_u$ is then represented as a sequence of semantic IDs: $\left[k_1^{(1)}, \ldots, k_N^{(1)}, \ldots, k_1^{(T_u)}, \ldots, k_N^{(T_u)}\right]$, which serves as input to train a Transformer model (Vaswani et al., 2017). To enable conditioning on natural language, we leverage pre-trained language encoders. Specifically, we represent both the interaction history and the user preference in natural language and process them with the pre-trained encoder. This is inspired by Li et al. (2023); Paischer et al. (2022; 2023), who demonstrated the benefits of history compression via language models. The decoder of Mender is randomly initialized and conditioned on the language encoder via cross-attention to predict semantic IDs.

We propose two variants of Mender, namely Mender_Tok and Mender_Emb. The key difference between these variants lies in how they encode user preferences and items. Mender_Tok encodes user preferences and items as a single sequence of language tokens. In contrast, Mender_Emb encodes each user preference and item separately using a pre-trained embedding model from Su et al. (2023). Mender_Emb allows pre-computing item and preference embeddings, resulting in improved training efficacy. Mender_Emb does not support fine-tuning, as propagating through the embedding model for each preference/item is prohibitively expensive. However, Mender_Tok processes the entire token sequence at once, making it amenable to fine-tuning.

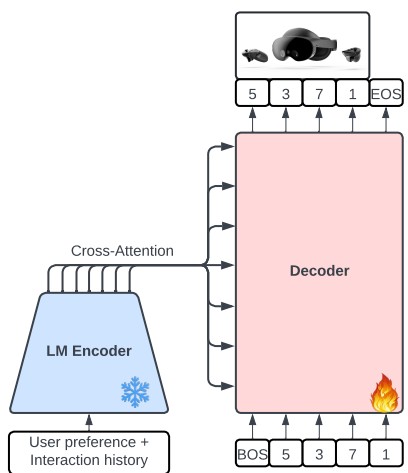

Figure 2: Architecture of Mender. The decoder generates semantic IDs conditioned on user preferences and past interactions via cross-attention with a pretrained language encoder.

## 4 EXPERIMENTS

We evaluate our approach on four widely-used datasets, namely three Amazon reviews subsets (Ni et al., 2019) and Steam (Kang & McAuley, 2018). An overview of the dataset statistics can be found in Table 3 in Appendix B. To generate user preferences, we utilize the `LlaMa-3-70B-Instruct`[1] model. For sentiment classification of reviews, we employ the model trained by Hartmann et al. (2023)[2]. The resulting preference statistics, including the number of generated preferences, the proportion of positive and negative preferences, and the sample sizes for each evaluation split, are presented in Table 4. Our data generation pipeline is built entirely on open-source models, making it easily extensible to additional datasets.

For training our models, we use the preference-based recommendation data, which consists of a single user preference and the interaction history. Unless mentioned otherwise, the additional generated data splits (positive/negative and fine/coarse data) are used solely for evaluation purposes. Following (Rajput et al., 2023), we limit the maximum number of items in a user sequence to the 20 most recent ones. We adopt a leave-last-out data split, where the penultimate item in a sequence is used for validation and the last item is used for testing (Kang & McAuley, 2018; Sun et al., 2019). Our evaluation benchmark is based only on validation and test items of that split. The remaining items in each sequence are used for training, except for the first item, since no user preferences are available for it. We evaluate our trained baselines using common retrieval metrics, including Recall (or Hit Rate), and Normalized Discounted Cumulative Gain (Järvelin & Kekäläinen, 2002, NDCG). Implementation details for training the RQ-VAE and Transformer models can be found in Appendix A.1 and Appendix A.2, respectively. All our methods are trained on single A100 or V100 GPUs using PyTorch (Paszke et al., 2019).

### 4.1 BASELINES

We train and evaluate a range of generative retrieval baselines and compare their performance to our Mender variants on our proposed benchmarks.

---

[1] https://huggingface.co/meta-llama/Meta-Llama-3-70B-Instruct

[2] https://huggingface.co/siebert/sentiment-roberta-large-english

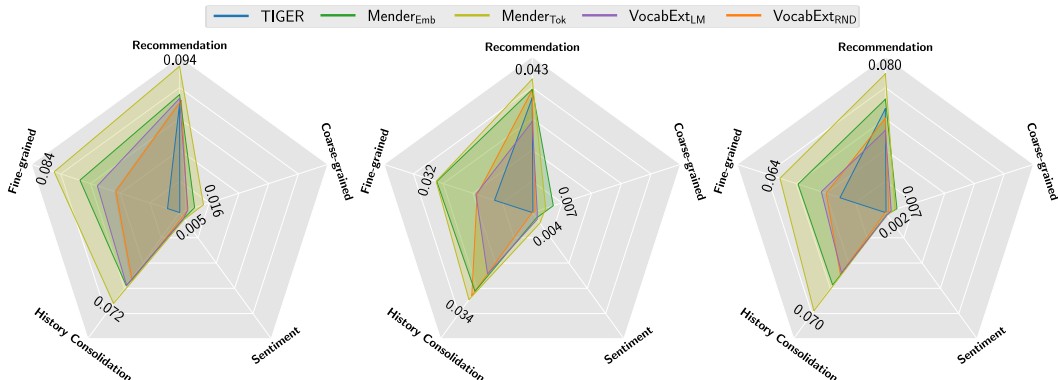

Figure 3: Recall@10 for all methods on our novel benchmark, evaluating preference discerning across three subsets of the Amazon review dataset: Beauty (left), Sports and Outdoors (middle), and Toys and Games (right). The different axes represent *recommendation*, *fine–grained steering*, *coarse-grained steering*, *history consolidation*, and *sentiment following*.

**TIGER** (Rajput et al., 2023) is a state-of-the-art generative retrieval model based on semantic IDs. Although TIGER is not conditioned on user preferences, we still evaluate its performance on our benchmarks for recommendation, fine-grained steering, and coarse-grained steering. The latter two essentially evaluate how well TIGER predicts a very similar or distinct item to the ground truth item. **VocabExt$_{RND}$** is based on extending the vocabulary of the TIGER model, which enables conditioning it on language preferences. Notably, this version does not leverage any pre-trained components. **LC-REC** (Zheng et al., 2023) extends the vocabulary of a pre-trained LM with newly initialized embeddings that represent semantic IDs. We fine-tune the LM utilizing LoRA (Hu et al., 2022), but do not add the auxiliary tasks. Additionally, we reduce the dimensionality of the language model head to match the number of semantic IDs, as language generation is not required for our task. **VocabExt$_{LM}$** represents the past interaction history in language as done for Mender$_{Tok}$ and Mender$_{Emb}$, but initializes the decoder with a pre-trained language decoder. Therefore this baseline operates on the same semantic gap as Mender variants. We again leverage LoRA for fine-tuning.

### 4.2 RESULTS

We present a detailed analysis of the results obtained by the different methods on our benchmark for three subsets of Amazon reviews (Beauty, Sports and Outdoors, and Toys and Games), and Steam datasets. Figure 3 and Figure 4 (left) show Recall@10 for all methods on the Amazon and Steam datasets, respectively. Table 1 shows complementary metrics, such as NDCG@5, NDCG@10, and Recall@5, as well as relative improvements of Mender to the best baseline method. In Appendix E we report the corresponding standard deviations for all methods across all datasets. Our results reveal several key trends: (i) incorporating preferences consistently improves performance; (ii) training on preference-based recommendation data leads to the emergence of fine-grained steering on certain datasets; (iii) current models struggle with sentiment following; and (iv) both coarse-grained steering and sentiment following can be achieved through data augmentations. Additionally, we provide ablation studies on data mixtures and the impact of adding user preferences in Section 4.3.

**Recommendation.** Our Mender$_{Tok}$ achieves the best performance across all datasets on the recommendation axis with relative improvements of up to 45%. The significant gap between TIGER and Mender$_{Tok}$ demonstrates the benefits of conditioning on the generated user preferences. The higher performance on the Steam dataset compared to the Amazon datasets traces back to the different item distributions (see Figure 8). As there are few items that are overrepresented, it is generally easier to obtain a higher score by predicting only those. Further, Mender$_{Emb}$ performs second-best on the Amazon datasets providing a decent trade-off between performance and training speed, by training around five times faster. Notably, other baselines such as VocabExt$_{RND}$ and LC-REC sometimes perform worse than TIGER on Toys and Steam, indicating that they cannot properly align the semantic id and language spaces. LC-REC usually requires auxiliary tasks to align the two spaces properly (Zheng et al., 2023), while our Mender successfully fuses them without training on auxiliary tasks.

Table 1: Performance for all methods on all evaluation axes for all datasets trained on recommendation data. We report average performance across three random seeds as well as relative improvements of Mender to the second-best performing baseline and highlight best performance in boldface. For sentiment following we reprt $m@k$ for $k \in \{5, 10\}$ instead of Recall@k.

| Methods | Sports and Outdoors | | | | Beauty | | | | Toys and Games | | | | Steam | | | |
|---|---|---|---|---|---|---|---|---|---|---|---|---|---|---|---|---|
| | Recall@5 | NDCG@5 | Recall@10 | NDCG@10 | Recall@5 | NDCG@5 | Recall@10 | NDCG@10 | Recall@5 | NDCG@5 | Recall@10 | NDCG@10 | Recall@5 | NDCG@5 | Recall@10 | NDCG@10 |
| Recommendation | | | | | | | | | | | | | | | | |
| TIGER | 0.0249 | 0.0158 | 0.0377 | 0.0199 | 0.0431 | 0.0275 | 0.0681 | 0.0356 | 0.0375 | 0.0238 | 0.0600 | 0.0311 | 0.163 | **0.144** | 0.193 | 0.153 |
| VocabExt$_{RND}$ | 0.0238 | 0.0151 | 0.0392 | 0.0201 | 0.0434 | 0.0277 | 0.0697 | 0.0362 | 0.0330 | 0.0205 | 0.0544 | 0.0275 | 0.166 | 0.142 | 0.200 | 0.154 |
| LC-REC | 0.0195 | 0.0124 | 0.0291 | 0.0156 | 0.0457 | 0.0294 | 0.0731 | 0.0382 | 0.0327 | 0.0209 | 0.0473 | 0.0256 | 0.160 | 0.137 | 0.194 | 0.148 |
| VocabExt$_{LM}$ | 0.0233 | 0.0148 | 0.0355 | 0.0187 | 0.0345 | 0.0224 | 0.0561 | 0.0293 | 0.0371 | 0.0234 | 0.0559 | 0.0296 | 0.1547 | 0.1305 | 0.1878 | 0.1412 |
| Mender$_{Emb}$ | 0.0264 | 0.0173 | 0.0394 | 0.0215 | 0.0494 | 0.0321 | 0.0755 | 0.0405 | 0.0422 | 0.0267 | 0.0653 | 0.0342 | 0.145 | 0.111 | 0.182 | 0.123 |
| Mender$_{Tok}$ | **0.0282** | **0.0188** | **0.0427** | **0.0234** | **0.0605** | **0.0401** | **0.0937** | **0.0508** | **0.0533** | **0.0346** | **0.0799** | **0.0432** | **0.168** | **0.144** | **0.204** | **0.156** |
| Rel. Impr. | +13.2% | +18.9% | +8.9% | +16.4% | +32.4% | +36.4% | +28.1% | +33.0% | +42.1% | +45.4% | +33.2% | +38.9% | +1.2% | +0.0% | +2.0% | +1.3% |
| Fine-grained steering | | | | | | | | | | | | | | | | |
| TIGER | 0.0061 | 0.0037 | 0.0118 | 0.0055 | 0.0119 | 0.0074 | 0.0195 | 0.0098 | 0.0149 | 0.0092 | 0.0237 | 0.0120 | 0.0084 | 0.0052 | 0.0145 | 0.0072 |
| VocabExt$_{RND}$ | 0.0104 | 0.0063 | 0.0186 | 0.0089 | 0.0229 | 0.0163 | 0.0437 | 0.0220 | 0.0200 | 0.0123 | 0.0358 | 0.0174 | 0.0102 | 0.0064 | 0.0178 | 0.0088 |
| LC-REC | 0.0119 | 0.0074 | 0.0190 | 0.0097 | 0.0348 | 0.0218 | 0.0563 | 0.0288 | 0.0248 | 0.0153 | 0.0388 | 0.0198 | 0.0157 | 0.0098 | 0.0264 | 0.0133 |
| VocabExt$_{LM}$ | **0.0214** | **0.0132** | **0.0352** | **0.0176** | 0.0292 | 0.0186 | 0.0498 | 0.0253 | 0.0341 | 0.0220 | 0.0572 | 0.0294 | **0.0217** | 0.0133 | **0.0365** | **0.0180** |
| Mender$_{Emb}$ | 0.0173 | 0.0106 | 0.0322 | 0.0154 | 0.0276 | 0.0174 | 0.0465 | 0.0234 | 0.0316 | 0.0199 | 0.0529 | 0.0267 | 0.0184 | 0.0114 | 0.0287 | 0.0147 |
| Mender$_{Tok}$ | 0.0190 | 0.0117 | 0.0324 | 0.0159 | **0.0534** | **0.0344** | **0.0844** | **0.0444** | **0.0378** | **0.0237** | **0.0639** | **0.0321** | 0.0211 | **0.0134** | 0.0352 | 0.0179 |
| Rel. Impr. | -12.6% | -12.8% | -8.6% | -10.7% | +53.4% | +57.8% | +49.9% | +54.2% | +10.9% | +7.7% | +11.7% | +9.2% | -2.8% | +1% | -3.7% | -1% |
| Coarse-grained steering | | | | | | | | | | | | | | | | |
| TIGER | 0.0001 | 0.0000 | 0.0003 | 0.0001 | 0.0003 | 0.0001 | 0.0003 | 0.0002 | 0.0003 | 0.0001 | 0.0006 | 0.0003 | 0.0005 | 0.0003 | 0.0008 | 0.0004 |
| VocabExt$_{RND}$ | 0.0005 | 0.0003 | 0.0010 | 0.0004 | 0.0023 | 0.0014 | 0.0046 | 0.0021 | 0.0013 | 0.0009 | 0.0021 | 0.0011 | 0.0032 | 0.0018 | 0.0055 | 0.0026 |
| LC-REC | 0.0010 | 0.0006 | 0.0017 | 0.0009 | 0.0032 | 0.0019 | 0.0059 | 0.0028 | 0.0022 | 0.0013 | 0.0036 | 0.0017 | 0.0028 | 0.0018 | 0.0049 | 0.0024 |
| VocabExt$_{LM}$ | **0.0047** | **0.0028** | **0.0098** | **0.0044** | 0.0053 | 0.0033 | 0.0086 | 0.0044 | **0.0037** | **0.0022** | 0.0065 | 0.0030 | **0.0047** | **0.0029** | 0.0077 | 0.0039 |
| Mender$_{Emb}$ | 0.0036 | 0.0022 | 0.0071 | 0.0033 | 0.0057 | 0.0035 | 0.0101 | 0.0050 | 0.0035 | 0.0021 | **0.0071** | **0.0032** | 0.0042 | 0.0024 | 0.0067 | 0.0032 |
| Mender$_{Tok}$ | 0.0023 | 0.0013 | 0.0045 | 0.0021 | **0.0094** | **0.0059** | **0.0161** | **0.0080** | 0.0032 | 0.0020 | 0.0060 | 0.0029 | 0.0043 | 0.0027 | **0.0081** | **0.0040** |
| Rel. Impr. | -30.6% | -27.3% | -38.1% | -33.3% | +77.4% | +78.8% | +87.2% | +81.8% | -15.6% | -4.8% | +9.2% | +6.7% | -9.3% | -7.4% | +5.2% | +2.6% |
| Sentiment following | | | | | | | | | | | | | | | | |
| TIGER | 0.0000 | - | 0.0000 | - | 0.0000 | - | 0.0000 | - | 0.0000 | - | 0.0000 | - | 0.0000 | - | 0.0000 | - |
| VocabExt$_{RND}$ | 0.0000 | - | 0.0000 | - | 0.0000 | - | 0.0000 | - | 0.0000 | - | 0.0000 | - | 0.0061 | - | 0.0086 | - |
| LC-REC | 0.0018 | - | 0.0027 | - | 0.0029 | - | 0.0045 | - | 0.0008 | - | **0.0017** | - | 0.0033 | - | 0.0053 | - |
| VocabExt$_{LM}$ | 0.0019 | - | 0.0016 | - | 0.0027 | - | 0.0051 | - | 0.0012 | - | 0.0004 | - | 0.0049 | - | 0.0107 | - |
| Mender$_{Emb}$ | 0.0022 | - | 0.0022 | - | 0.0030 | - | 0.0047 | - | **0.0017** | - | 0.0015 | - | **0.0114** | - | **0.0185** | - |
| Mender$_{Tok}$ | **0.0035** | - | **0.0042** | - | **0.0043** | - | **0.0053** | - | 0.0016 | - | **0.0017** | - | 0.0084 | - | 0.0110 | - |
| Rel. Impr. | +84.2% | - | +55.6% | - | +48.3% | - | +3.9% | - | +41.7% | - | +0% | - | +86.9% | - | +72.9% | - |
| History consolidation | | | | | | | | | | | | | | | | |
| TIGER | 0.0000 | 0.0000 | 0.0000 | 0.0000 | 0.0000 | 0.0000 | 0.0000 | 0.0000 | 0.0000 | 0.0000 | 0.0000 | 0.0000 | 0.0000 | 0.0000 | 0.0000 | 0.0000 |
| VocabExt$_{RND}$ | 0.0190 | 0.0120 | 0.0329 | 0.0164 | 0.0303 | 0.0191 | 0.0504 | 0.0256 | 0.0260 | 0.0158 | 0.0441 | 0.0216 | 0.1366 | 0.1155 | 0.1642 | 0.1244 |
| LC-REC | 0.0158 | 0.0101 | 0.0243 | 0.0129 | 0.0354 | 0.0226 | 0.0577 | 0.0297 | 0.0295 | 0.0185 | 0.0430 | 0.0229 | **0.1460** | **0.1277** | **0.1726** | **0.1363** |
| VocabExt$_{LM}$ | 0.0179 | 0.0112 | 0.0278 | 0.0145 | 0.0247 | 0.0155 | 0.0423 | 0.0211 | 0.0316 | 0.0195 | 0.0487 | 0.0251 | 0.0615 | 0.0440 | 0.0866 | 0.0521 |
| Mender$_{Emb}$ | 0.0206 | 0.0133 | 0.0312 | 0.0167 | 0.0352 | 0.0228 | 0.0580 | 0.0301 | 0.0314 | 0.0201 | 0.0516 | 0.0266 | 0.1241 | 0.0938 | 0.1558 | 0.1040 |
| Mender$_{Tok}$ | **0.0234** | **0.0151** | **0.0345** | **0.0187** | **0.0457** | **0.0304** | **0.0720** | **0.0388** | **0.0467** | **0.0302** | **0.0700** | **0.0377** | 0.0490 | 0.0317 | 0.0745 | 0.0399 |
| Rel. Impr. | +23.2% | +25.8% | +4.9% | +14.0% | +29.1% | +34.5% | +24.8% | +30.6% | +58.3% | +54.9% | +43.7% | +50.2% | -15.1% | -26.5% | -9.7% | -23.7% |

VocabExt$_{RND}$ performs significantly worse then both Mender versions due to its lack of a pre-trained language encoder, which requires learning the interaction between item history and user preferences from scratch. In contrast, LC-REC utilizes a pre-trained language encoder but fails to effectively combine semantic IDs with language representations. A potential reason for this is that our datasets are relatively small and only comprise recommendation data and no grounding of semantic IDs to language. Based on these findings, we conclude that: (i) user preferences substantially enhance recommendation performance, and (ii) representing both interaction history and user preferences in a linguistic format is the preferred approach to fuse interaction history and generated user preferences.

**Fine- and coarse-grained steering.** We observe that Mender$_{Tok}$ consistently achieves the best performance across all datasets for fine-grained steering with relative improvements of up to 70.5% to baselines. Notably, as illustrated in Figure 3, fine-grained steering naturally emerges as a byproduct of training on preference-based recommendation data. However, this is not the case for the Steam dataset (Figure 4, right), where we notice a significant gap between recommendation and fine-grained steering performance. We surmise that the reason for this is the fundamental difference in the respective data distribution of the Amazon and Steam datasets. Prior work demonstrated that data distribution is an essential driving factor to elicit emerging capabilites such as in-context learning (Chan et al., 2022). Future work should aim at confirming this conjecture via systematic experiments. Furthermore, our results indicate that all methods struggle to perform coarse-grained

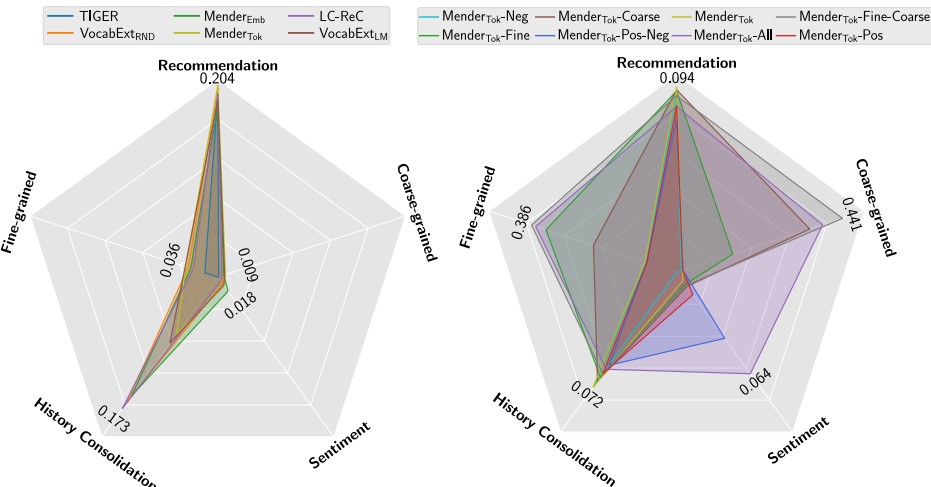

Figure 4: Recall@10 of different baselines trained on the default recommendation data of the Steam dataset (left). Recall@10 for Mender$_{\text{Tok}}$ trained on different datasplits on the Amazon Beauty subset, evaluated under various schemes: *Recommendation*, *Sentiment following*, *Preference steering*, *Preference consolidation*, and *History consolidation* (right).

steering, suggesting that the preference-based recommendation data lacks a beneficial signal to facilitate the emergence of coarse-grained steering.

**History Consolidation.** Generally, we observe that all methods attain lower scores on history consolidation compared to the recommendation. This is because the additional preferences are not necessarily related to the ground truth item and thus add a substantial amount of noise. Furthermore, one of the five user preferences provided to the model contains information to identify the ground truth item as they were matched during the benchmark generation. Therefore, the attained performance is a proxy on how well the model can identify a useful preference out of set of potentially noisy preferences. On the Amazon subsets, Mender$_{\text{Tok}}$ consistently attains the highest performance, while LC-REC attains the best results on Steam. These findings suggest that preference-based methods can effectively fuse interaction history with multiple user preferences for recommendation. Interestingly, we found in preliminary experiments that training on history consolidation data alone leads to a degradation in recommendation performance. Contrary, our reported results demonstrate that training on preference-based recommendation data maintains recommendation performance and elicits decent performance on history consolidation *and* recommendation.

**Sentiment Following.** While both Mender variants attain the highest performance on different datasets, the overall performance on sentiment following is generally around an order of magnitude smaller. This result indicates that all current models struggle with sentiment following. This finding presents an interesting avenue for future research, which should prioritize developing models that can accurately identify the sentiment of user preferences and adapt their retrieval accordingly. Prior works found that there is little to no gain by incorporating negative user preferences into recommendation systems (Sanner et al., 2023). Our results confirm that current systems mostly lack the ability to discern negative preferences and to act accordingly. However, we show in the next section that this observation depends on *how* the negative preferences are used during training, and that it is indeed possible to obtain a system that improves along this axis.

### 4.3 ABLATION STUDIES

**Importance of Preferences.** We perform an ablation study to investigate the impact of combining both user preferences and items represented in natural language. In Figure 7 in Appendix A.4 we provide evidence that representing items in language instead of semantic IDs leads to better rankings. Further, we quantify the improvement by providing both, preferences along with items represented in language in the model's context. To this end, we train Mender$_{\text{Tok}}$ and (i), condition it only on preferences; (ii) condition it only on items represented in language; and (iii), condition it on both.

We present our results for the Beauty dataset in Figure 5, right. Our results clearly demonstrate the benefits of combining items with user preferences in language.

**Data Mixture.** We evaluate whether models trained on sentiment following and steering improve performance on the respective evaluation axes. This is particularly interesting for datasets, such as Steam, where no steering capabilities emerged, or the Amazon subsets, where models lack coarse-grained steering. We augment the training set with the additional data sources and train different variants of $\text{Mender}_{\text{Tok}}$. We train four variants: $\text{Mender}_{\text{Tok}}$-Pos, which uses positive pairs; $\text{Mender}_{\text{Tok}}$-Neg, which uses negative pairs; $\text{Mender}_{\text{Tok}}$-Pos-Neg, which combines both positive and negative pairs; $\text{Mender}_{\text{Tok}}$-Fine, which uses fine-grained steering data; $\text{Mender}_{\text{Tok}}$-Coarse, which uses coarse-grained steering data; $\text{Mender}_{\text{Tok}}$-Fine-Coarse, which uses fine- and coarse-grained steering data; and finally, $\text{Mender}_{\text{Tok}}$-All, which is trained on all data. When including the negative $(p_u^-, i)$ tuples, we simply minimize the likelihood and weight it by a hyperparameter. We present Recall@10 for Beauty in Figure 4, right

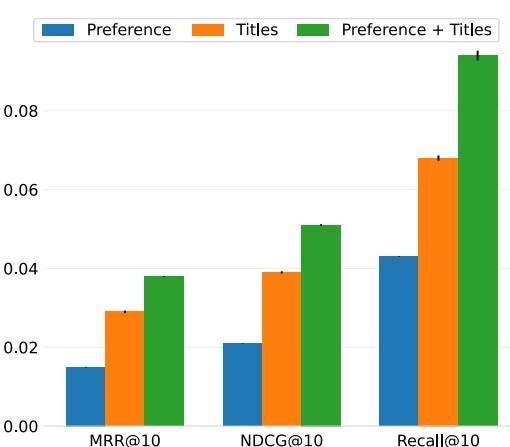

Figure 5: **Left: Right:** Ablation study highlighting the importance of combining items and user preferences in natural language.

and for Steam in Appendix E. We additionally report Recall@5, NDCG@5, and NDCG@10 for all methods and evaluation axes in Table 5 in Appendix E. Most importantly, coarse-grained steering as well as sentiment following capabilities arise when we explicitly train the model on the respective data. Interestingly, $\text{Mender}_{\text{Tok}}$-All significantly improves upon $\text{Mender}_{\text{Tok}}$ on all axes while maintaining performance on the recommendation axis. However, training on a data split in isolation improves over training on all data, i.e. $\text{Mender}_{\text{Tok}}$-Coarse leads to better coarse-grained steering than $\text{Mender}_{\text{Tok}}$-All, but lacks sentiment following. Furthermore, sentiment following capabilities only arise when training on both, positive and negative data jointly. These findings present a fruitful avenue for future research on mixing the different data sources.

## 5 LIMITATIONS

A current limitation of our benchmark is that the compiled user preferences are limited to five selected datasets. However, since we used open-source models to generate them, the data generation pipeline can be extended to new datasets. Currently, the data generation process relies on extensive post-processing to ensure high-quality user preferences, which is tailored to the specific LLM we used. Furthermore, our preference generation pipeline relies on the presence of user reviews and does not take into account longer time dependencies. Finally, we do not explore the effect of scaling the language encoder. All of these limitations present fruitful avenues for future work.

## 6 CONCLUSION

Current sequential recommendation systems are limited in their personalization as they model user preferences implicitly. We propose a new paradigm, namely preference discerning, in which the sequential recommendation system is conditioned on user preferences represented in natural language. To evaluate preference discerning capabilities, we present a benchmark that is specifically designed to assess the ability of sequential recommendation models to discern textual preferences along five different axes. We also propose a novel generative retrieval model, Mender, which represents the items at different levels of abstraction, namely semantic ids *and* natural language. Our experimental results show that Mender outperforms state-of-the-art models on our benchmark. Our contributions pave the way for a new class of generative retrieval models that unlock leveraging organic data for steering via textual user preferences.

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

# SUPPLEMENTARY MATERIAL

**Anonymous authors**

## CONTENTS

## A  GENERATIVE RETRIEVAL VIA SEMANTIC IDS

We provide an open-source implementation of all baselines used in this work, including TIGER (Rajput et al., 2023). To facilitate reproducibility of the results reported in Rajput et al. (2023), we elaborate on the implementation details as follows. The training of TIGER consists of two stages: (i) training the residual quantizer (RQ-VAE) to obtain semantic IDs, and (ii), training the generative retrieval model.

### A.1  RQ-VAE

Training the RQ-VAE involves two essential steps: (i) constructing an item embedding, and (ii) optimizing the model through residual quantization.

**Item embedding** For item embedding, we utilize the Sentence-T5 model (Ni et al., 2022), which is publicly available on the Hugging Face Hub (Wolf et al., 2020). We explored various sources of information to represent items and found that the optimal approach varies across datasets. For the Beauty and Sports datasets, using item descriptions led to sub-optimal results due to the high noise levels present in these descriptions. In contrast, item descriptions proved beneficial for the Toys dataset. Additionally, we leveraged other item attributes, including title, price, brand, and categories. For the Stream dataset, we utilized a broader set of attributes: title, genre, specs, tags, price, publisher, and sentiment.

**Training** By default, we standardize the item embeddings, as this helps prevent collapse during RQ-VAE training. For training the RQ-VAE, we found that the encoder-decoder architecture is crucial for increasing codebook coverage. Additionally, residual connections and weight decay

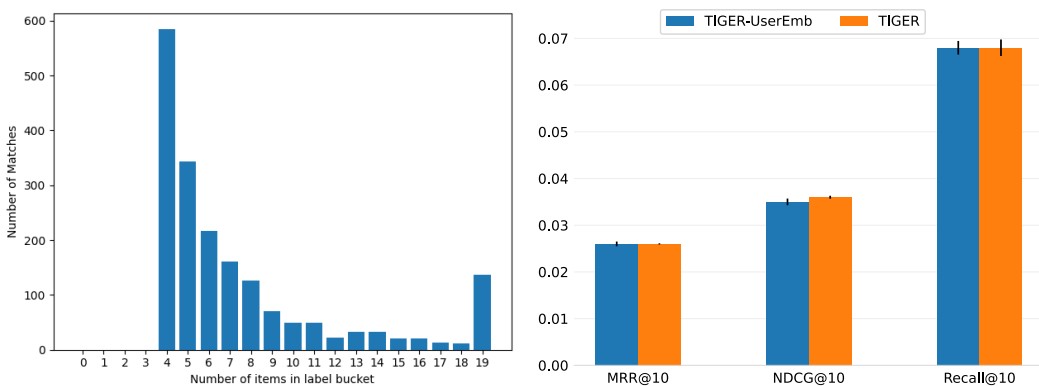

Figure 6: **Left:** Number of correctly retrieved test items for TIGER on the Beauty subset of the Amazon review dataset. **Right:** Performance comparison of TIGER with user embedding (TIGER-UserEmb) and without user embedding (TIGER) on the Beauty dataset.

are essential for maintaining a good separation and preventing collapse. Our encoder architecture consists of four hidden layers with sizes 768, 512, 256, and 128, respectively. Each layer includes layer normalization (Ba et al., 2016), ReLU activation, and dropout (Hinton et al., 2012). The decoder follows the same architecture but in reverse order, where the sum of residuals obtained via the quantization procedure is up-projected to the original dimension of 768. Following Rajput et al. (2023), we use a three-level residual quantization scheme with 256 codebooks each. We also experimented with EMA updates and resetting unused codebook entries, as in Lee et al. (2022), but did not observe any significant improvements. To evaluate the performance of our trained RQ-VAEs, we rely on metrics such as reconstruction error, codebook coverage, and downstream task performance.

### A.2 TRANSFORMER

Following Rajput et al. (2023) we instantiate the generative model via the T5 architecture (Raffel et al., 2020). Next, we delve into the design choices underlying this approach, as introduced by Rajput et al. (2023), and discuss their utility.

**Training sequences** To construct the training sequences, Rajput et al. (2023) limit the number of items in a user sequence to at most 20. This can be implemented by taking either the first, the last, or all items within a sliding window of up to 20 items. We experimented with each of these approaches and found that using the most recent 20 items in a user sequence generally yields improved performance. Unlike prior sequential recommendation systems, which require at least one item in a sequence to predict the next item (Kang & McAuley, 2018; Zhou et al., 2020), TIGER leverages a user embedding trained alongside item embeddings. Therefore, we typically use the first item in a sequence for training, although this is not strictly necessary.

**Decoding** Another crucial aspect of the generative retrieval pipeline is the decoding process. As noted by Rajput et al. (2023), the generation of valid semantic IDs is not guaranteed. To mitigate this issue, we track the number of invalid semantic IDs produced during decoding. We find that this number is typically quite low. Nevertheless, to further improve the accuracy of our retrieval results, we employ filtering to remove invalid IDs and increase the beam size to be larger than the final retrieval set.

### A.3 REPRODUCED RESULTS

In Table 2, we present the results of our reproduced and open-source implementation of TIGER. Our results closely match those reported in Rajput et al. (2023) for the Sports and Beauty datasets, but we observe a significant gap on the Toys dataset. Notably, our trained models achieve substantially higher Recall@10 scores on the Beauty dataset. Furthermore, we find that the disparity is more pro-

nounced for NDCG than for Recall, suggesting that while the retrieved candidate items are similar, our models' ranking performance is inferior.

Table 2: Reproduced results for our open-source implementation of TIGER (Rajput et al., 2023)

| Methods | Sports and Outdoors | | | | Beauty | | | | Toys and Games | | | |
|---|---|---|---|---|---|---|---|---|---|---|---|---|
| | Recall @5 | NDCG @5 | Recall @10 | NDCG @10 | Recall @5 | NDCG @5 | Recall @10 | NDCG @10 | Recall @5 | NDCG @5 | Recall @10 | NDCG @10 |
| P5 Geng et al. (2022) | 0.0061 | 0.0041 | 0.0095 | 0.0052 | 0.0163 | 0.0107 | 0.0254 | 0.0136 | 0.0070 | 0.0050 | 0.0121 | 0.0066 |
| Caser Tang & Wang (2018) | 0.0116 | 0.0072 | 0.0194 | 0.0097 | 0.0205 | 0.0131 | 0.0347 | 0.0176 | 0.0166 | 0.0107 | 0.0270 | 0.0141 |
| HGN Ma et al. (2019) | 0.0189 | 0.0120 | 0.0313 | 0.0159 | 0.0325 | 0.0206 | 0.0512 | 0.0266 | 0.0321 | 0.0221 | 0.0497 | 0.0277 |
| GRU4Rec Hidasi et al. (2016a) | 0.0129 | 0.0086 | 0.0204 | 0.0110 | 0.0164 | 0.0099 | 0.0283 | 0.0137 | 0.0097 | 0.0059 | 0.0176 | 0.0084 |
| BERT4Rec Sun et al. (2019) | 0.0115 | 0.0075 | 0.0191 | 0.0099 | 0.0203 | 0.0124 | 0.0347 | 0.0170 | 0.0116 | 0.0071 | 0.0203 | 0.0099 |
| FDSA Zhang et al. (2019b) | 0.0182 | 0.0122 | 0.0288 | 0.0156 | 0.0267 | 0.0163 | 0.0407 | 0.0208 | 0.0228 | 0.0140 | 0.0381 | 0.0189 |
| SASRec Kang & McAuley (2018) | 0.0233 | 0.0154 | 0.0350 | 0.0192 | 0.0387 | 0.0249 | 0.0605 | 0.0318 | 0.0463 | 0.0306 | 0.0675 | 0.0374 |
| $S^3$-Rec Zhou et al. (2020) | 0.0251 | 0.0161 | 0.0385 | 0.0204 | 0.0387 | 0.0244 | 0.0647 | 0.0327 | 0.0443 | 0.0294 | 0.0700 | 0.0376 |
| **TIGER(Rajput et al., 2023)** | 0.0264 | 0.0181 | 0.0400 | 0.0225 | 0.0454 | 0.0321 | 0.0648 | 0.0384 | 0.0521 | 0.0371 | 0.0712 | 0.0432 |
| **TIGER (Ours)** | 0.0249 | 0.0158 | 0.0377 | 0.0199 | 0.0431 | 0.0275 | 0.0681 | 0.0356 | 0.0375 | 0.0238 | 0.0600 | 0.0311 |

## A.4 ADDITIONAL FINDINGS

Beyond the experiments discussed above, we conducted further investigations into the TIGER framework, yielding the following key insights.

- TIGER exhibits superior performance on shorter sequences, as shown in Figure 6 (left).
- The inclusion of user embeddings in TIGER does not yield any significant benefits to downstream performance, as illustrated in Figure 6 (right).
- Representing interaction history in natural language leads to improved ranking performance, as demonstrated in Figure 7.

**TIGER Works Better on Shorter Sequences.** As shown in Figure 6 (left), TIGER performs significantly better on shorter sequences than on longer ones. The x-axis represents the number of items per test sequence, which is at least 4 due to the 5-core user and item filtering applied. Further, the maximum number of items per sequence is capped at 19, as we limit the maximum sequences length to 20, following (Rajput et al., 2023). This results in a maximum sequence length of 19 items, where the task is to predict the 20th item. The y-axis shows the number of matches. Notably, TIGER's performance is substantially better on shorter sequences than on longer ones. However, the number of matches increases again for the longest sequences, although it remains considerably lower than for shorter sequences.

**User Embedding.** Rajput et al. (2023) employ a user embedding selected based on hashing. However, it is unclear whether this approach offers any advantages, as the number of user embeddings suggested by Rajput et al. (2023) often results in numerous collisions in practice. To investigate this, we conduct an experiment where we remove the user embedding entirely. As shown in Figure 6 (middle), we do not observe a significant drop in performance. This suggests that the user embedding, as implemented in (Rajput et al., 2023), does not provide any notable benefits.

**History Compression via Natural Language.** We conduct an additional study where we represent the past interaction history using natural language and initialize the encoder of TIGER with a FLAN-T5-small language encoder. This approach is reminiscent of history compression via language models (Paischer et al., 2022, HELM). We refer to this variant as LIGER (Language-TIGER), and compare its performance to the baseline TIGER in Figure 7. The results show that while there is no significant difference in Recall, LIGER yields notable improvement in NDCG metrics. This suggests that compressing interaction history using natural language generally enhances the model's ranking capabilities.

## B DATASETS

We consider two publicly available datasets for sequential recommendation: Amazon review dataset (Ni et al., 2019) and Steam (Kang & McAuley, 2018). To preprocess these datasets, we apply a 5-core filtering criterion, removing users with fewer than five interactions and items that appear

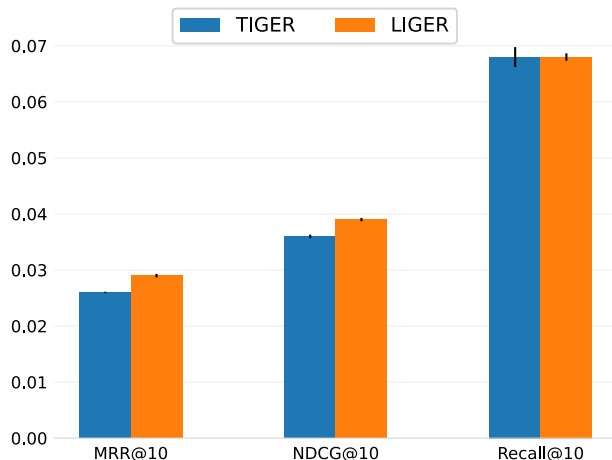

Figure 7: Performance comparison between TIGER and LIGER on the Beauty subset of the Amazon review dataset. Both models predict semantic IDs, but differ in their input representation: LIGER encodes past items as natural language descriptions, while TIGER represents them as semantic IDs.

less than five times. The resulting dataset statistics are presented in Table 3. Due to computational constraints, we subsample the Steam dataset to reduce the number of user preferences generated during the preference approximation pipeline.

We also visualize the item distribution in Figure 8, which shows that the three Amazon datasets follow approximatelx the same item distribution, while for Steam the distribution differs significantly. In particular, on the Steam dataset the number of items in the same range as for the Amazon datasets, however the number of users is much larger, as well as the average number of actions per user. As can be observed from the item distribution, there is a small fraction of items that is overrepresented.

Table 3: Dataset statistics after user 5-core and item 5-core preprocessing. Asterisk denotes datasets are subsets of the Amazon review dataset.

| Dataset | #users | #items | avg. actions /user | avg. actions /item | #actions |
|---|---|---|---|---|---|
| *Beauty\** | 22,363 | 12,101 | 8.8764 | 16.403 | 198,502 |
| *Toys and Games\** | 19,412 | 11,924 | 8.6337 | 14.0554 | 167,597 |
| *Sports and Outdoors\** | 35,598 | 18,357 | 8.3245 | 16.1430 | 296,337 |
| *Yelp* | 19,855 | 14,540 | 10.4279 | 14.2387 | 207,045 |
| *Steam* | 47,761 | 10,403 | 12.554 | 54.6549 | 599,620 |

## C   PREFERENCE GENERATION

In this section, we provide details on the prompting scheme used to generate user preferences from item reviews using `LLaMA-3-70B-Instruct`. We provide reviews along with item specific information to the LLM and prompt it to generate a set of five user preferences (see Figure 9). Below we present an example prompt and response for a user in the Beauty subset of the Amazon reviews dataset.

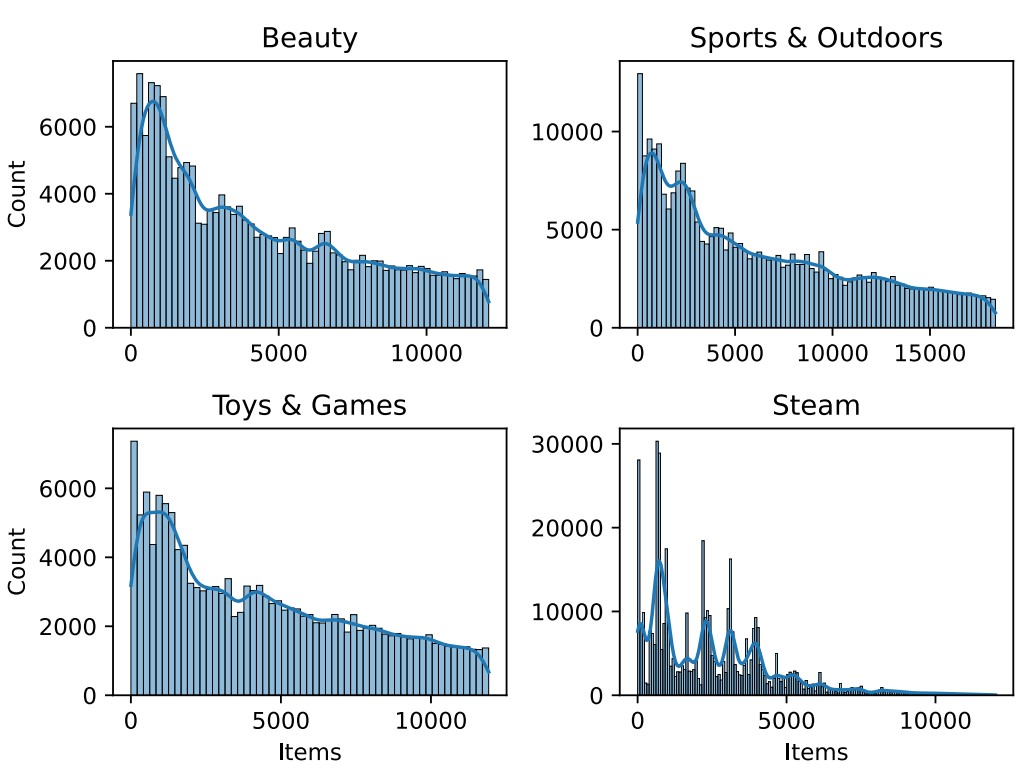

Figure 8: Data distribution of the Amazon and Steam datasets.

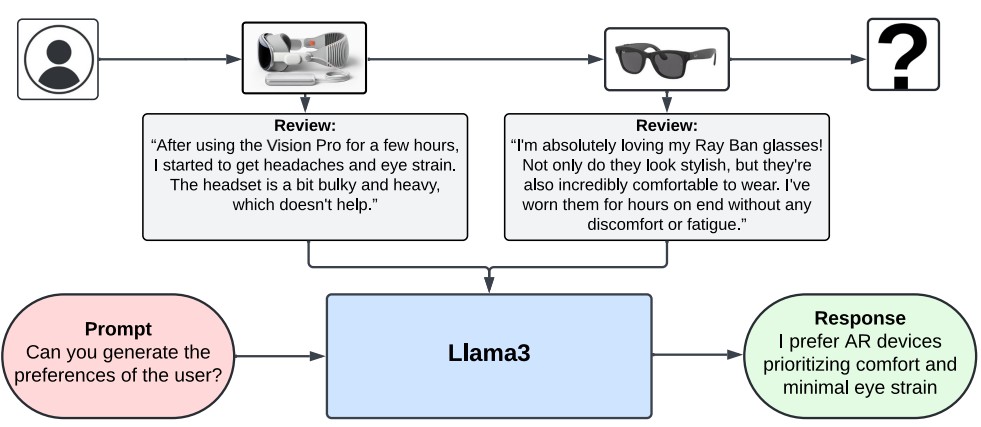

Figure 9: Schematic illustration of our preference generation pipeline. A user's reviews for items, combined with item information, are input into Llama3 as a prompt to infer the user's preferences.

Table 4: Statistics for generated preferences for the different datasets. For pos/neg and fine/coarse we show number of samples in the format train/val/test split.

| Benchmark | #preferences | #positive | #negative | pos/neg | fine/coarse |
|---|---|---|---|---|---|
| *Beauty* | 992,510 | 708,706 | 283,804 | 17,811/3,671/3,716 | 24,114/16,702/15,956 |
| *Toys and Games* | 837,985 | 645,696 | 192,289 | 11,513/2,342/2,508 | 23,730/15,968/14,950 |
| *Sports and Outdoors* | 1,481,685 | 1,075,679 | 406,006 | 21,402/4,275/4,293 | 36,552/25,728/25,188 |
| *Steam* | 2,026,225 | 1,495,931 | 530,294 | 31,505/7,968/8,493 | 19,550/10,678/10,626 |

---

**Instruction:**
Here is a list of items a user bought along with their respective reviews in json format:    { } . Your task is to generate a list of up to five search instructions that reflect the user's preferences based on their reviews. Be specific about what the user likes, does not like, and should be avoided. Do not mention brands or certain products. Return a json file containing the search instructions with key 'instructions'. Keep the instructions simple, short and concise, and do NOT include comments on delivery time or pricing.

**Parsed response:**
```
['Search for nail polish with shimmer finish', 'Look for products
with vibrant, bold colors', 'Avoid products that require base coat
for optimal results', 'Prioritize products with high-quality,
long-lasting formula', 'Opt for products with easy, smooth
application']
```

---

After generation, we apply an exhaustive postprocessing step to ensure that every user-item pair is associated with exactly five user preferences. In Table 4 we show the statistics after our preference generation pipeline for the different datasets.

**Granularity of preferences.** We also investigate whether the granularity of user preferences affects the model's ability to adhere to them. We experiment with various prompts to elicit preferences at different levels of granularity, ranging from coarser, higher-level user preferences to fine-grained preferences that include detailed descriptions of products and brands. To obtain more abstract user preferences, we slightly modify the prompt as follows. An example for the same user mentioned above is shown.

---

**Instruction:**
Here is a list of items a user bought along with their respective reviews in json format:    { } . Your task is to generate a list of up to five search instructions that summarizes the users high level preferences based on their reviews. Be specific on what the user does not like and should be avoided. Do not mention brands or certain products. Return a json file containing the search instructions with key 'instructions'. Keep the instructions simple, short and concise, and do NOT include comments on delivery time or pricing.

**Parsed response:**
```
['Look for products with vibrant colors', 'Prioritize products with
high-quality finishes', 'Opt for products that are easy to apply',
'Consider products that are suitable for multiple uses (e.g.,
toes and fingers)', 'Avoid products that require additional base
coats']
```

---

Finally, to obtain more fine-grained user preferences, we slightly modify the prompt again to permit the inclusion of specific brands and item names in the LLama's response. As shown below, the LLama indeed mentions brands when given the explicit instruction that mentioning item names or brands is acceptable.

> **Instruction:**
> Here is a list of items a user bought along with their respective reviews in json format:   { } .
> Your task is to generate a list of up to five search instructions that reflect the user's preferences based
> on their reviews. Be specific on what the user likes, does not like, and should be avoided. It is ok
> to mention brands or certain products. Return a json file containing the search instructions with key
> 'instructions'. Keep the instructions simple, short and concise, and do NOT include comments on
> delivery time or pricing.
> **Parsed response:**
> *["Search for nail polishes with similar shimmering effects to*
> *OPI's Simmer and Shimmer", 'Prioritize products with vibrant,*
> *long-lasting colors that look great on toes', "Look for nail*
> *polish brands that offer a wide range of colors, similar to*
> *OPI's Burlesque collection", 'Avoid products that require a base*
> *coat for optimal results', 'Opt for nail polishes with a smooth,*
> *easy-to-apply formula']*

In practice, we found that varying the granularity of user preferences did not significantly impact model performance when using more fine/coarse grained user preferences. This suggests that the user preferences obtained through our default prompting scheme contain sufficient information to effectively steer the model.

## C.1 REVIEWS TO PROPERTIES

In addition to item titles and descriptions, the collective reviews of items offer a wealth of information about their properties. However, with hundreds of users potentially reviewing a single item, each contributing multiple paragraphs, extracting relevant information can be challenging. To improve the signal-to-noise ratio, we prompt `LLaMA-3-70B-Instruct` to extract and condense item properties from the corpus of user reviews. Specifically, we instruct the LLM to provide concise objective descriptions, excluding subjective opinions. Below, we illustrate this approach with an example of extracting properties from reviews of a froggy pool in the Amazon Toys and Games dataset.

> **Instruction:**
> Your task is to summarize the following reviews of an item into a list of item properties using
> keywords and phrases:   { } . Keep your response short and concise. Only focus on objective
> properties of the item. Do NOT include subjective opinions or emotions. Do NOT include comments
> on price or delivery time. Return your response as a python list with at most 10 entries that accurately
> reflect the properties of the item.
> **Parsed response:**
> *Inflatable pool with a frog head design, Rainbow sunshade provides*
> *shade for the entire pool, Inflatable base consists of multiple air*
> *cushions, Pool holds 12 gallons of water, Maximum fill line is 4*
> *inches, Inflated pool area has a 28 inch diameter, Three separate*
> *air valves for inflation, Easy to set up and take down, Suitable*
> *for children aged 1-3 years, Cushioned inflatable base for comfort.*
> **Original review:**
> *This froggy pool is great for your young children.  It is 45" x 39"*
> *x 27" and very comfortable and safe for your child.*

The parsed response above demonstrates that LLama3 is capable of providing a comprehensive list of characteristics about toys in the Amazon Toys and Games dataset. We observed modest performance gains when providing summarized item properties alongside item titles and preferences as input to our language-conditioned generative retrieval model. Since we were limited by the number of prompts we could issue, we did not compile item properties for all datasets. Nevertheless, we provide the prompts to facilitate future research in this direction.

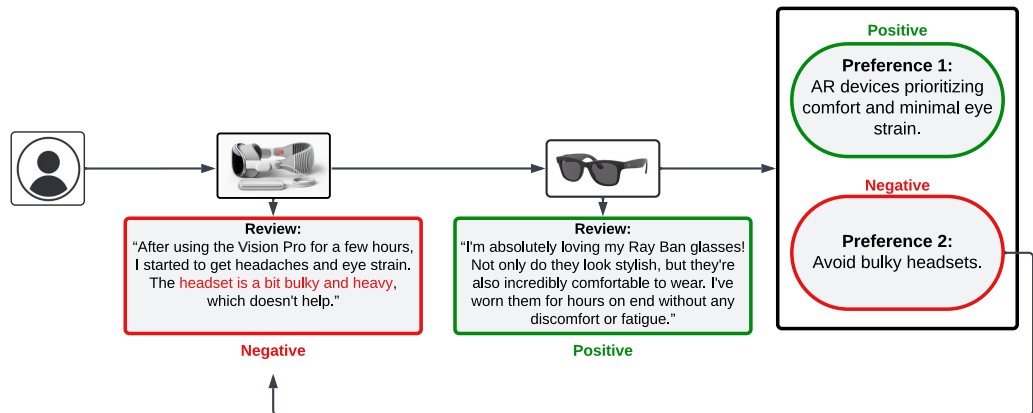

Figure 10: Schematic illustration of our pipeline to identify the reviews that triggered negative user preferences. The reviews of different items guided the LLM to generate two distinct user preferences. We perform sentiment classification on both user preferences and reviews, followed by a matching step in Sentence-T5 space to determine which negative review led to a negative user preference.

## D   BENCHMARK DESIGN

In this section, we provide supplementary detail on the creation of the various components of our benchmark.

### D.1   PREFERENCE SENTIMENT UNDERSTANDING

The sentiment understanding benchmark is based on preference-item pairs and utilizes a matching mechanism to identify items that triggered negative reviews. This is implemented using a pre-trained sentiment classification model from Hartmann et al. (2023) to classify reviews. To identify preferences, we employ a rule-based approach, as we observed that preferences can be both positive and negative simultaneously (e.g., a preference may specify liking certain items, while avoiding others). Furthermore, we noticed that negative preferences consistently follow a specific pattern, starting with either "*Avoid*", "*Exclude*", or "*No*". To minimize misclassifications, we consider preferences beginning with these words to be negative. If only one item in a user sequence received a negative review, we pair the negative preference with that item. Otherwise, we use a matching mechanism in Sentence-T5 space, where we match a negative preference to the item whose review is closest in terms of cosine similarity. An example of the negative matching pipeline is illustrated in Figure 10 . This yields a set of negative preference-item pairs, enabling us to evaluate whether the model can recognize negative sentiment and respond accordingly. To obtain positive preference-item pairs, we iterate over all negative pairs and invert the gathered preferences. Since negative instructions always start with "*Avoid*", "*Exclude*", or "*No*", we simply replace these words with "*Find*" or "*Search for*" to invert them. This results in two sets: one containing negative preferences paired with items and another containing positive preferences paired with the same items. Finally, we assess whether the model can successfully avoid certain items, while actively retrieving others.

### D.2   PREFERENCE STEERING

In the preference steering scenario we consider two distinct scenarios: *fine-grained* and *coarse-grained* preference steering. The former assesses whether the model can retrieve an item very similar to the ground truth by modifying the user preference. In contrast, the latter evaluates whether the model can retrieve a distinctly different item by changing the user preference accordingly. We identify a very similar item by the maximal cosine similarity in a pre-trained Sentence-T5 embedding space. Contrary, we retrieve a very distinct item by the lowest cosine similarity to the ground truth item. Subsequently, we match the retrieved items to new user preferences, again via cosine

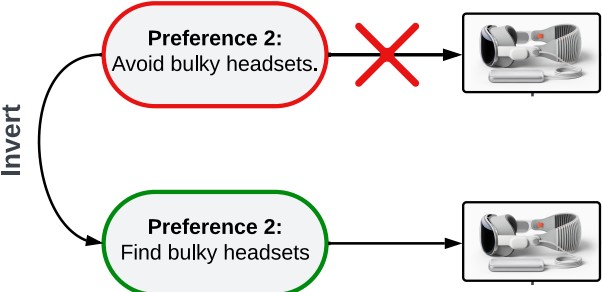

Figure 11: Positive and negative preference-item pairs obtained after matching negative preferences to items that received a negative review. We apply a rule-based inversion to generate the corresponding positive pair.

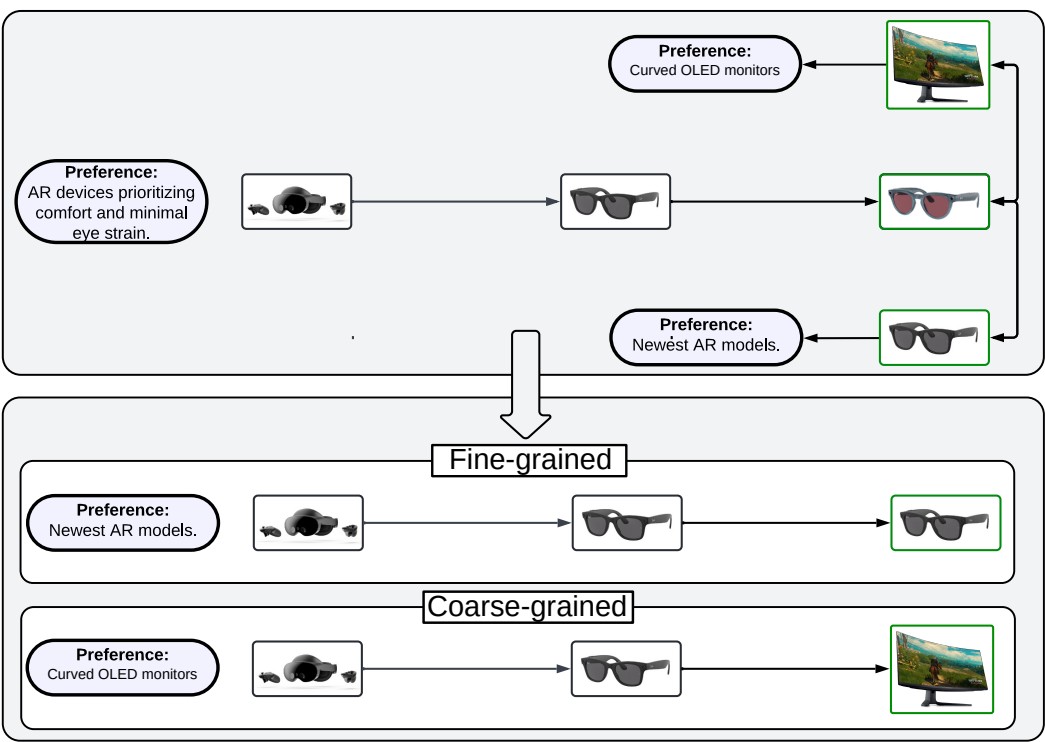

Figure 12: Schematic illustration of our pipeline for constructing fine- and coarse-grained preference steering. We search for very similar and dissimilar items to the ground truth item of each original item sequence and match them to user preferences (top). Then, we obtain two new sequences by exchanging the original preference with each user preferences and associated new ground truth item.

similarity. We show a visual illustration of this procedure in Figure 12. Finally, we ensure that there is no overlap between our compiled training, validation and test split by controlling for the matched preferecnes, i.e. if a user preference was already matched to a retrieved item, we associate the current item with the next most similar or distinct preference. This results in uniquely (preference, item) tuples for every dataset split.

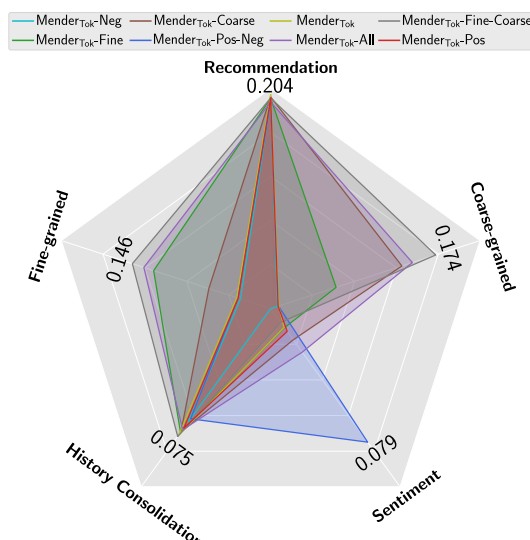

Figure 13: Recall@10 for Mender$_{Tok}$ trained on different datasplits on the Steam dataset, evaluated under various schemes: *Recommendation*, *Sentiment following*, *Preference steering*, *Preference consolidation*, and *History consolidation*.

# E  ADDITIONAL RESULTS

We provide complementary results for our ablation studies on the data mixture. In Table 5 we report Recall@5, Recall@10, NDCG@5, and NDCG@10 for the different versions of Mender that are trained on different data mixes. Furthermore, we provide results for training on the Steam dataset with different data mixtures in Figure 13 to highlight that fine-and coarse-grained steering as well as sentiment following capabilties can be obtained on this dataset as well.

Additionally, we report standard deviations of our results in Table 1 in Table 6 with larger values being depicted in red. The small standard deviation indicates that reported improvements of Mender are statistically significant.

To assess the efficiency of our Mender variants, we compare the time required for training and inference as well as their performance. Furthermore, we add a comparison to SASRec (Kang & McAuley, 2018) which is a traditional sequential recommendation baseline. We present our results in Table 7 for the all the four datasets.

Additionally, we conduct an experiment to demonstrate that training on all five generated user preferences leads to detrimental performance. As mentioned in Section 3.2, each training sequence contains of a single user preference that is matched to the target item in a pre-trained SentenceT5 space. To verify that this is the best training strategy, we compare Mender$_{Tok}$ trained on these sequences to the setup where Mender$_{Tok}$ receives all five user preferences along with the interaction history (Mender$_{Tok}$-AllPrefs), i.e. the training sequences are structured as $\left[p_{u_1}^{T_u-1}, \ldots, p_{u_5}^{T_u-1}, i_1, \ldots, i_{T_u-1}\right]$. We report our results in Table 8. They verify that training on sequences $\left[p_u^{T_u-1}, \ldots, i_1, \ldots, i_{T_u-1}\right]$ where $p_u^{T_u-1}$ is matched to the ground truth item $i_{T_u-1}$ attains significantly better results no recommendation and also leads to the emergence of steering capabilities.

Table 5: Performance for different versions of Mender trained on different data mixtures for all evaluation axes on the Beauty and Steam datasets. We report average performance across three random seeds.

| Methods | Beauty | | | | Steam | | | |
|---|---|---|---|---|---|---|---|---|
| | Recall @5 | NDCG @5 | Recall @10 | NDCG @10 | Recall @5 | NDCG @5 | Recall @10 | NDCG @10 |
| Recommendation | | | | | | | | |
| $Mender_{Tok}$ | 0.0605 | 0.0401 | 0.0937 | 0.0508 | 0.1682 | 0.1441 | 0.2037 | 0.1555 |
| $Mender_{Tok}$-Pos | 0.0553 | 0.0371 | 0.0840 | 0.0463 | 0.1667 | 0.1429 | 0.2004 | 0.1538 |
| $Mender_{Tok}$-Neg | 0.0598 | 0.0394 | 0.0917 | 0.0497 | 0.1646 | 0.1410 | 0.1983 | 0.1519 |
| $Mender_{Tok}$-Pos-Neg | 0.0491 | 0.0321 | 0.0778 | 0.0413 | 0.1647 | 0.1416 | 0.1979 | 0.1523 |
| $Mender_{Tok}$-Fine | 0.0591 | 0.0383 | 0.0918 | 0.0487 | 0.1667 | 0.1428 | 0.2005 | 0.1538 |
| $Mender_{Tok}$-Coarse | 0.0601 | 0.0392 | 0.0924 | 0.0496 | 0.1682 | 0.1440 | 0.2018 | 0.1549 |
| $Mender_{Tok}$-Fine-Coarse | 0.0570 | 0.0366 | 0.0893 | 0.0470 | 0.1663 | 0.1424 | 0.2007 | 0.1535 |
| $Mender_{Tok}$-All | 0.0529 | 0.0337 | 0.0838 | 0.0436 | 0.1634 | 0.1400 | 0.1969 | 0.1508 |
| Fine-grained steering | | | | | | | | |
| $Mender_{Tok}$ | 0.0534 | 0.0344 | 0.0844 | 0.0444 | 0.0218 | 0.0137 | 0.0357 | 0.0182 |
| $Mender_{Tok}$-Pos | 0.0501 | 0.0321 | 0.0791 | 0.0414 | 0.0217 | 0.0137 | 0.0343 | 0.0177 |
| $Mender_{Tok}$-Neg | 0.0500 | 0.0323 | 0.0803 | 0.0420 | 0.0196 | 0.0124 | 0.0318 | 0.0163 |
| $Mender_{Tok}$-Pos-Neg | 0.0513 | 0.0333 | 0.0791 | 0.0423 | 0.0211 | 0.0131 | 0.0344 | 0.0173 |
| $Mender_{Tok}$-Fine | 0.2476 | 0.1680 | 0.3475 | 0.2002 | 0.0829 | 0.0538 | 0.1234 | 0.0668 |
| $Mender_{Tok}$-Coarse | 0.1483 | 0.0981 | 0.2212 | 0.1215 | 0.0395 | 0.0244 | 0.0652 | 0.0327 |
| $Mender_{Tok}$-Fine-Coarse | 0.2781 | 0.1885 | 0.3861 | 0.2234 | 0.0985 | 0.0643 | 0.1459 | 0.0795 |
| $Mender_{Tok}$-All | 0.2676 | 0.1802 | 0.3750 | 0.2148 | 0.0903 | 0.0601 | 0.1338 | 0.0741 |
| Coarse-grained steering | | | | | | | | |
| $Mender_{Tok}$ | 0.0094 | 0.0059 | 0.0161 | 0.0080 | 0.0045 | 0.0028 | 0.0085 | 0.0041 |
| $Mender_{Tok}$-Pos | 0.0098 | 0.0062 | 0.0163 | 0.0083 | 0.0047 | 0.0029 | 0.0079 | 0.0040 |
| $Mender_{Tok}$-Neg | 0.0063 | 0.0039 | 0.0117 | 0.0056 | 0.0041 | 0.0027 | 0.0072 | 0.0036 |
| $Mender_{Tok}$-Pos-Neg | 0.0095 | 0.0061 | 0.0169 | 0.0084 | 0.0050 | 0.0031 | 0.0083 | 0.0041 |
| $Mender_{Tok}$-Fine | 0.1005 | 0.0655 | 0.1494 | 0.0813 | 0.0272 | 0.0175 | 0.0691 | 0.0304 |
| $Mender_{Tok}$-Coarse | 0.3028 | 0.2631 | 0.3541 | 0.2797 | 0.0953 | 0.0485 | 0.1385 | 0.0624 |
| $Mender_{Tok}$-Fine-Coarse | 0.3525 | 0.2710 | 0.4413 | 0.2999 | 0.1403 | 0.1052 | 0.1741 | 0.1163 |
| $Mender_{Tok}$-All | 0.3294 | 0.2779 | 0.3885 | 0.2970 | 0.1063 | 0.0696 | 0.1495 | 0.0839 |
| Sentiment following | | | | | | | | |
| $Mender_{Tok}$ | 0.0043 | - | 0.0053 | - | 0.0084 | - | 0.0110 | - |
| $Mender_{Tok}$-Pos | 0.0113 | - | 0.0140 | - | 0.0123 | - | 0.0134 | - |
| $Mender_{Tok}$-Neg | 0.0000 | - | 0.0000 | - | 0.0000 | - | 0.0000 | - |
| $Mender_{Tok}$-Pos-Neg | 0.0268 | - | 0.0414 | - | 0.0637 | - | 0.0787 | - |
| $Mender_{Tok}$-Fine | 0.0046 | - | 0.0075 | - | 0.0080 | - | 0.0112 | - |
| $Mender_{Tok}$-Coarse | 0.0067 | - | 0.0089 | - | 0.0088 | - | 0.0184 | - |
| $Mender_{Tok}$-Fine-Coarse | 0.0057 | - | 0.0083 | - | 0.0053 | - | 0.0081 | - |
| $Mender_{Tok}$-All | 0.0440 | - | 0.0635 | - | 0.0184 | - | 0.0256 | - |
| History consolidation | | | | | | | | |
| $Mender_{Tok}$ | 0.0457 | 0.0304 | 0.0720 | 0.0388 | 0.0490 | 0.0317 | 0.0745 | 0.0399 |
| $Mender_{Tok}$-Pos | 0.0405 | 0.0272 | 0.0632 | 0.0344 | 0.0490 | 0.0331 | 0.0704 | 0.0400 |
| $Mender_{Tok}$-Neg | 0.0460 | 0.0301 | 0.0714 | 0.0383 | 0.0448 | 0.0288 | 0.0667 | 0.0359 |
| $Mender_{Tok}$-Pos-Neg | 0.0359 | 0.0233 | 0.0581 | 0.0305 | 0.0440 | 0.0293 | 0.0649 | 0.0360 |
| $Mender_{Tok}$-Fine | 0.0418 | 0.0270 | 0.0657 | 0.0346 | 0.0492 | 0.0333 | 0.0730 | 0.0410 |
| $Mender_{Tok}$-Coarse | 0.0436 | 0.0284 | 0.0682 | 0.0363 | 0.0495 | 0.0331 | 0.0728 | 0.0406 |
| $Mender_{Tok}$-Fine-Coarse | 0.0399 | 0.0254 | 0.0636 | 0.0331 | 0.0517 | 0.0355 | 0.0753 | 0.0430 |
| $Mender_{Tok}$-All | 0.0379 | 0.0236 | 0.0607 | 0.0309 | 0.0506 | 0.0349 | 0.0713 | 0.0416 |

# F  USER STUDY

Our aim is to verify that the user preferences that were generated by the LLM accurately approximate the real user preferences. To this end, we conduct a user study to answer the following questions:

Table 6: Standard deviation for all methods on all evaluation axes for all datasets trained on recommendation data across three random seeds.

| Methods | Sports and Outdoors | | | | Beauty | | | | Toys and Games | | | | Steam | | | |
|---|---|---|---|---|---|---|---|---|---|---|---|---|---|---|---|---|
| | Recall@5 | NDCG@5 | Recall@10 | NDCG@10 | Recall@5 | NDCG@5 | Recall@10 | NDCG@10 | Recall@5 | NDCG@5 | Recall@10 | NDCG@10 | Recall@5 | NDCG@5 | Recall@10 | NDCG@10 |
| Recommendation | | | | | | | | | | | | | | | | |
| TIGER | 0.0009 | 0.0006 | 0.0006 | 0.0005 | 0.0010 | 0.0009 | 0.0012 | 0.0009 | 0.0008 | 0.0005 | 0.0004 | 0.0004 | 0.0015 | 0.0014 | 0.0008 | 0.0012 |
| VocabExt$_{RND}$ | 0.0002 | 0.0001 | 0.0002 | 0.0000 | 0.0020 | 0.0017 | 0.0034 | 0.0022 | 0.0005 | 0.0006 | 0.0006 | 0.0006 | 0.0006 | 0.0002 | 0.0015 | 0.0001 |
| LC-REC | 0.0021 | 0.0014 | 0.0027 | 0.0016 | 0.0010 | 0.0007 | 0.0006 | 0.0006 | 0.0010 | 0.0009 | 0.0015 | 0.0010 | 0.0014 | 0.0019 | 0.0013 | 0.0019 |
| Mender$_{Emb}$ | 0.0011 | 0.0005 | 0.0017 | 0.0007 | 0.0007 | 0.0007 | 0.0017 | 0.0010 | 0.0015 | 0.0010 | 0.0023 | 0.0012 | 0.0035 | 0.0030 | 0.0040 | 0.0031 |
| Mender$_{Tok}$ | 0.0007 | 0.0005 | 0.0005 | 0.0004 | 0.0004 | 0.0001 | 0.0012 | 0.0002 | 0.0019 | 0.0011 | 0.0022 | 0.0012 | 0.0006 | 0.0004 | 0.0004 | 0.0003 |
| Fine-grained steering | | | | | | | | | | | | | | | | |
| TIGER | 0.0006 | 0.0004 | 0.0006 | 0.0004 | 0.0040 | 0.0024 | 0.0065 | 0.0032 | 0.0010 | 0.0006 | 0.0032 | 0.0011 | 0.0005 | 0.0003 | 0.0010 | 0.0004 |
| VocabExt$_{RND}$ | 0.0007 | 0.0005 | 0.0006 | 0.0005 | 0.0005 | 0.0004 | 0.0019 | 0.0009 | 0.0009 | 0.0004 | 0.0010 | 0.0004 | 0.0010 | 0.0005 | 0.0011 | 0.0004 |
| LC-REC | 0.0034 | 0.0022 | 0.0054 | 0.0028 | 0.0009 | 0.0004 | 0.0018 | 0.0007 | 0.0016 | 0.0010 | 0.0024 | 0.0012 | 0.0014 | 0.0006 | 0.0020 | 0.0007 |
| Mender$_{Emb}$ | 0.0009 | 0.0005 | 0.0013 | 0.0007 | 0.0017 | 0.0013 | 0.0015 | 0.0012 | 0.0020 | 0.0017 | 0.0015 | 0.0015 | 0.0024 | 0.0014 | 0.0039 | 0.0019 |
| Mender$_{Tok}$ | 0.0004 | 0.0000 | 0.0010 | 0.0003 | 0.0012 | 0.0007 | 0.0007 | 0.0004 | 0.0006 | 0.0008 | 0.0004 | 0.0010 | 0.0005 | 0.0003 | 0.0004 | 0.0003 |
| Coarse-grained steering | | | | | | | | | | | | | | | | |
| TIGER | 0.0000 | 0.0000 | 0.0000 | 0.0000 | 0.0001 | 0.0000 | 0.0001 | 0.0001 | 0.0001 | 0.0001 | 0.0001 | 0.0001 | 0.0001 | 0.0001 | 0.0002 | 0.0001 |
| VocabExt$_{RND}$ | 0.0001 | 0.0000 | 0.0001 | 0.0000 | 0.0003 | 0.0002 | 0.0002 | 0.0000 | 0.0004 | 0.0003 | 0.0002 | 0.0002 | 0.0002 | 0.0001 | 0.0004 | 0.0001 |
| LC-REC | 0.0005 | 0.0003 | 0.0008 | 0.0004 | 0.0006 | 0.0003 | 0.0012 | 0.0005 | 0.0007 | 0.0005 | 0.0009 | 0.0005 | 0.0005 | 0.0004 | 0.0008 | 0.0004 |
| Mender$_{Emb}$ | 0.0000 | 0.0000 | 0.0004 | 0.0001 | 0.0008 | 0.0005 | 0.0000 | 0.0002 | 0.0009 | 0.0006 | 0.0009 | 0.0005 | 0.0005 | 0.0002 | 0.0010 | 0.0003 |
| Mender$_{Tok}$ | 0.0002 | 0.0001 | 0.0005 | 0.0002 | 0.0015 | 0.0011 | 0.0017 | 0.0011 | 0.0003 | 0.0002 | 0.0009 | 0.0004 | 0.0005 | 0.0003 | 0.0002 | 0.0001 |
| Sentiment following | | | | | | | | | | | | | | | | |
| TIGER | 0.0000 | - | 0.0000 | - | 0.0000 | - | 0.0000 | - | 0.0000 | - | 0.0000 | - | 0.0000 | - | 0.0000 | - |
| VocabExt$_{RND}$ | 0.0000 | - | 0.0000 | - | 0.0012 | - | 0.0005 | - | 0.0000 | - | 0.0000 | - | 0.0029 | - | 0.0010 | - |
| LC-REC | 0.0003 | - | 0.0007 | - | 0.0006 | - | 0.0012 | - | 0.0003 | - | 0.0007 | - | 0.0016 | - | 0.0014 | - |
| Mender$_{Emb}$ | 0.0001 | - | 0.0001 | - | 0.0003 | - | 0.0007 | - | 0.0002 | - | 0.0005 | - | 0.0003 | - | 0.0020 | - |
| Mender$_{Tok}$ | 0.0011 | - | 0.0012 | - | 0.0014 | - | 0.0003 | - | 0.0000 | - | 0.0002 | - | 0.0012 | - | 0.0014 | - |
| History consolidation | | | | | | | | | | | | | | | | |
| TIGER | 0.0000 | 0.0000 | 0.0000 | 0.0000 | 0.0000 | 0.0000 | 0.0000 | 0.0000 | 0.0000 | 0.0000 | 0.0000 | 0.0000 | 0.0000 | 0.0000 | 0.0000 | 0.0000 |
| VocabExt$_{RND}$ | 0.0001 | 0.0001 | 0.0007 | 0.0003 | 0.0017 | 0.0016 | 0.0020 | 0.0017 | 0.0009 | 0.0008 | 0.0006 | 0.0007 | 0.0023 | 0.0027 | 0.0028 | 0.0028 |
| LC-REC | 0.0009 | 0.0006 | 0.0012 | 0.0007 | 0.0012 | 0.0007 | 0.0012 | 0.0007 | 0.0008 | 0.0003 | 0.0018 | 0.0007 | 0.0014 | 0.0019 | 0.0012 | 0.0018 |
| Mender$_{Emb}$ | 0.0011 | 0.0005 | 0.0018 | 0.0007 | 0.0007 | 0.0003 | 0.0005 | 0.0002 | 0.0006 | 0.0008 | 0.0015 | 0.0007 | 0.0003 | 0.0007 | 0.0006 | 0.0008 |
| Mender$_{Tok}$ | 0.0008 | 0.0006 | 0.0007 | 0.0006 | 0.0005 | 0.0000 | 0.0005 | 0.0001 | 0.0015 | 0.0013 | 0.0014 | 0.0013 | 0.0030 | 0.0023 | 0.0038 | 0.0025 |

Table 7: Performance, training time and inference time on an A100 GPU for Mender$_{Emb}$, Mender$_{Tok}$, and traditional sequential recommendation system SASRec (Kang & McAuley, 2018) on Beauty, Sports and Outdoors, Toys and Games, and Steam.

| Method | Dataset | Train time | Inference time | NDGC@10 | Recall@10 |
|---|---|---|---|---|---|
| SASRec | Beauty | 293min | 8ms | $0.0227 \pm 0.0004$ | $0.0528 \pm 0.0006$ |
| | Sports and Outdoors | 447min | 9ms | $0.0118 \pm 0.0002$ | $0.0271 \pm 0.0005$ |
| | Toys and Games | 280min | 5ms | $0.0267 \pm 0.0002$ | $0.0615 \pm 0.0002$ |
| | Steam | 280min | 5ms | $0.1469 \pm 0.0002$ | $0.1781 \pm 0.0004$ |
| Mender$_{Emb}$ | Beauty | 127min | 453ms | $0.0405 \pm 0.001$ | $0.0755 \pm 0.0017$ |
| | Sports and Outdoors | 374min | 194ms | $0.0215 \pm 0.0007$ | $0.0394 \pm 0.0017$ |
| | Toys and Games | 239min | 178ms | $0.0342 \pm 0.0015$ | $0.0653 \pm 0.0015$ |
| | Steam | 231min | 179ms | $0.123 \pm 0.0031$ | $0.182 \pm 0.004$ |
| Mender$_{Tok}$ | Beauty | 2324min | 562ms | $0.0508 \pm 0.0002$ | $0.0937 \pm 0.0012$ |
| | Sports and Outdoors | 2350min | 210ms | $0.0234 \pm 0.0004$ | $0.0427 \pm 0.0005$ |
| | Toys and Games | 1021min | 227ms | $0.0432 \pm 0.0012$ | $0.0799 \pm 0.0022$ |
| | Steam | 2330min | 222ms | $0.156 \pm 0.0003$ | $0.204 \pm 0.0004$ |

1. Are the generated user preferences informed by the user's past interaction history?
2. Do the generated preferences accurately approximate the user's preferences?
3. Is the matched preference related to the target item?
4. Given that a user preference accurately approximates the user's preferences, is it related to the target item?

Table 8: Performance of Mender$_{\text{Tok}}$ when being trained on the single matched preference compared to training on all five generated user preferences on the Amazon datasets. For sentiment following we report $m@10$ instead of Recall@10.

| Methods | Beauty | | Sports | | Toys | |
|---|---|---|---|---|---|---|
| | Recall @10 | NDCG @10 | Recall @10 | NDCG @10 | Recall @10 | NDCG @10 |
| Recommendation | | | | | | |
| Mender$_{\text{Tok}}$ | 0.0937 | 0.0508 | 0.0427 | 0.0234 | 0.0799 | 0.0432 |
| Mender$_{\text{Tok}}$-AllPrefs | 0.0131 | 0.0066 | 0.0063 | 0.0037 | 0.0074 | 0.0039 |
| Fine-grained steering | | | | | | |
| Mender$_{\text{Tok}}$ | 0.0844 | 0.0444 | 0.0324 | 0.0159 | 0.0639 | 0.0321 |
| Mender$_{\text{Tok}}$-AllPrefs | 0.0014 | 0.0006 | 0.0009 | 0.0004 | 0.0018 | 0.0009 |
| Coarse-grained steering | | | | | | |
| Mender$_{\text{Tok}}$ | 0.0161 | 0.0080 | 0.0045 | 0.0021 | 0.0060 | 0.0029 |
| Mender$_{\text{Tok}}$-AllPrefs | 0.0006 | 0.0002 | 0.0003 | 0.0002 | 0.0006 | 0.0003 |
| Sentiment following | | | | | | |
| Mender$_{\text{Tok}}$ | 0.0053 | - | 0.0042 | - | 0.0017 | - |
| Mender$_{\text{Tok}}$-AllPrefs | 0.0008 | - | 0.0001 | - | 0.0005 | - |
| History consolidation | | | | | | |
| Mender$_{\text{Tok}}$ | 0.0720 | 0.0388 | 0.0345 | 0.0187 | 0.0700 | 0.0377 |
| Mender$_{\text{Tok}}$-AllPrefs | 0.0089 | 0.0041 | 0.0063 | 0.0038 | 0.0046 | 0.0025 |

In total there were 22 participants that answered all three aforementioned questions about 20 randomly sampled recommendation scenarios of one of the Beauty, Toys and Games, Sports and Outdoors, or Steam datasets. For each of the three questions we provide three possible answers, namely (1) yes, (2) no, or (3) lack of information to tell. In one such scenario, users were first shown the past interaction history of a random user along with their reviews. Then, the generated user preferences were displayed along with the one user preference that was matched to the ground truth item, i.e. the next item in the sequence. In the end, we also display the ground truth item with the same information as the recommendation system would receive it.

Overall, 440 recommendation scenarios were reviewed, which amounts to 2200 preferences that were judged. We now iterate over all questions and present the corresponding findings.

**Are the generated user preferences informed by the user's past interaction history?** The objective for introducing this question was to quantify how much of the generated preferences was actually represented in the interaction history and what amount has been *hallucinated*. We report the results for this first question in Figure 14. The majority of users found that the generated user preferences are generally well informed by the user's interaction history across datasets. We found that the model occasionally generated rather generic preferences, for example "Avoid harsh chemicals" on the Beauty dataset even though there was no mention of harsh datasets in the reviews. Such preferences are rather generic and do not convey much information about a user's preference. Furthermore, some participants indicated that there was a lack of information to answer the question. This can be traced back to the fact that we intentionally did not provide item descriptions to the user, as those often contain a substantial amount of noise. As this information is hidden, we believe it caused the small fraction of preferences that were rated as *lack of info*. Thus, we can conclude that the generated user preferences for the most part were informed by reviews or item-specific info, however there is still a non-negligible amount of user preferences that can be considered *hallucinated*.

**Do the generated preferences accurately approximate the user's preferences?** The aim of this question is to quantify whether participants believe that the preferences of the user are correctly

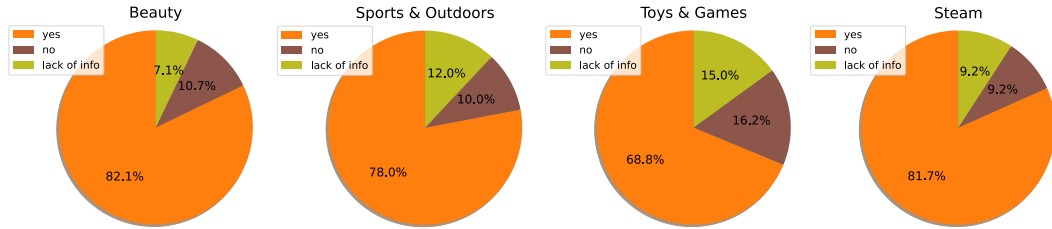

Figure 14: User survey results for the question "Are the generated user preferences informed by the user's past interaction history?" for the four different datasets used for approximating user preferences.

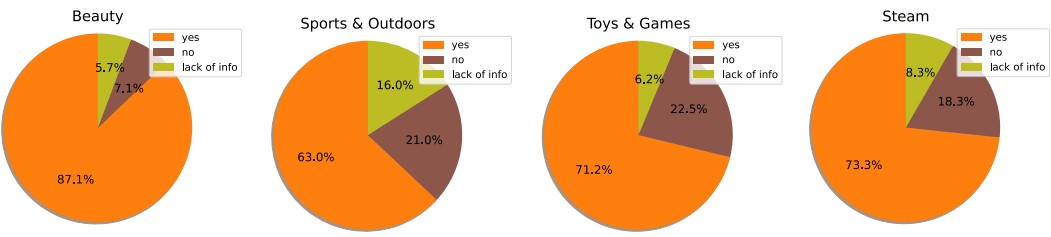

Figure 15: User survey results for the question "Do the generated preferences accurately approximate the user's preferences?" for the four different datasets used for approximating user preferences.

approximated. This question is crucial as it sits at the core of our user survey to identify the quality of the preferences. We report the outcome in Figure 15. Again we find that participants believe that for the most part, the preferences accurately reflects the user's preferences. In this case, the answer *lack of info* means that there is not enough information to capture the user's preferences, which is the case if very little details are given in the reviews or they are missing entirely. Fortunately, this case is underrepresented. Overall, we can conclude that the approximation of user preferences via our preference approximation results yields high quality preferences that accurately reflect the user's preferences.

**Is the matched preference related to the target item?** After we have established the quality of the preferences it is imperative to also evaluate our matching of preferences to target items conducted in Section 3.2 for preference-based recommendation. The reason we conduct this matching is to provide the model with a useful signal it can learn from. This is imperative as we observed that simply using all preferences for training leads to detrimental performance (see Table 8). We report the results for this question in Figure 16. Interestingly, the fraction of correctly matched preferences is significantly lower compared to the number of correctly generated preferences. The reasons for this can be two fold, (i) it can be that the target item is entirely unrelated to the past interaction history, or (ii), the matching mechanism is suboptimal. The former case reflects the inherent uncertainty of the sequential recommendation task, as oftentimes the target item is simply not related to previously acquired purchases. This shortcoming cannot be alleviated. The latter, however, can be tackled by potentially more expressive embedding models or LLMs that can be used for matching preferences to the target item. Finally, the *lack of info* category represents cases where the information about the target item is too little, i.e. no description or item title is given. Overall, we can conclude that even though we demonstrated significant performance gains resulting from training on the matched preferences, it could likely be improved.

**Given that a user preference accurately approximates the user's preferences, is it related to the target item?** This question was not explicitly asked for in the user study, however we can obtain an estimate on the underlying aleatoric uncertainty of the task. In particular, we consider cases where Q2 was answered with yes and visualize the three categories for Q3 (see Figure 17). In other words, we look at correctly approximated preferences and ask what fraction of them is related to the target

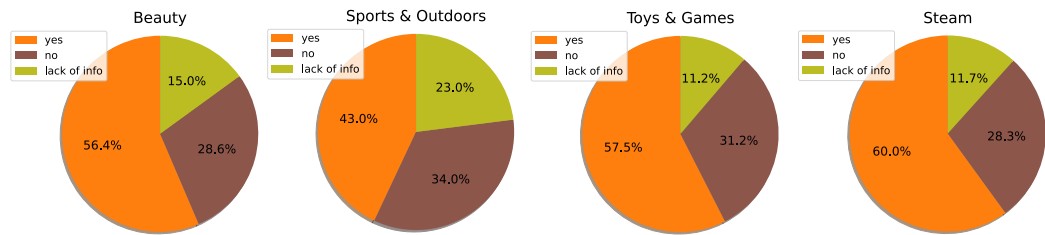

Figure 16: User survey results for the question "Is the matched preference related to the target item?" for the four different datasets used for approximating user preferences.

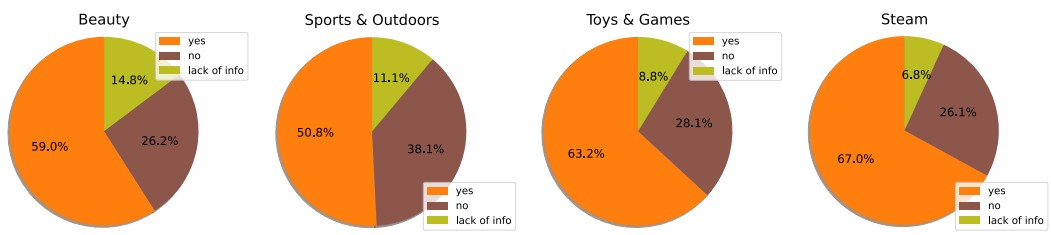

Figure 17: User survey results for the question 'Given that a user preference accurately approximates the user's preferences, is it related to the target item?" for the four different datasets used for approximating user preferences.

item. If Q2 is answered with *yes*, then we expect the matching to perform well, as there is a semantic relation to the target item. However, if there is still no relation to the target item, i.e. Q3 is answered with *no*, then we know this is due to the inherent uncertainty of the task. Interestingly, 50-70% of the correctly approximated preferences are related to the target item. This provides us with an upper bound on the maximum performance that can be obtained on the sequential recommendation task, i.e. the maximum Recall that can be obtained is in the range of 0.5-0.7, depending on the dataset.

