# OpenReview forum: "Preference Discerning in Generative Sequential Recommendation"
_ICLR.cc/2025/Conference — Submitted to ICLR 2025_

### Official Review · Reviewer_Fpwn · 2024-10-17

**Soundness:** 3
**Presentation:** 2
**Contribution:** 2
**Rating:** 3
**Confidence:** 4

**Summary:**

This paper first introduces a new benchmark to evaluate the model's ability to capture user preference. Then Mender is proposed to integrate LLM-generated user preferences to enhance the generative recommender system. Experiment results on its proposed benchmark show improvement on the proposed method.

**Strengths:**

1. The motivation of this work, leveraging user preference in recommender system, is good.
2. The author conducts extensive experiments.

**Weaknesses:**

Regarding Benchmark Design:

1. While preference-based recommendation is undoubtedly a core aspect, the practical value of the tasks such as Sentiment Following, Fine-Grained & Coarse-Grained Steering, and History Consolidation is questionable. This raises concerns about the overall contribution of the benchmark.
2. The Fine-Grained & Coarse-Grained Steering task is confusing. The paper states, “we associate these two items with different user preferences, denoted as p1 and p2, respectively,” but the relationship between p1, p2, and similar or distinct items is unclear. How are “different user preferences” determined? Additionally, in the new sequences created, why is p1 added to the sequence of very distinct items while p2 is added to the ground truth item sequence? This contradicts the earlier association of p1 and p2 with similar and distinct items, respectively. What role does the similar item play?
3. The design of the Sentiment Following task does not adequately reflect the model’s ability to follow sentiment. The description is also unclear, and I suggest the authors reorganize this section.
4. The practical value of History Consolidation is questionable, and its evaluation metric seems unnecessary. Why not train the model directly using the five preferences? The paper claims to “infer which preference is most relevant for predicting,” but there is no experimental evidence demonstrating this capability. In fact, the performance with multiple preferences is even worse than with a single preference.
5. The experimental discussion on each task, particularly Sentiment Following and History Consolidation, is insufficient.

Regarding Presentation:
1. Missing reference on line 1275.
2. Typo on line 1167: "tr iggered" should be "triggered."
3. Typo on line 401: "48.3.4%" should be corrected.
4. Method names are displayed incorrectly in lines 403-406.
5. In Table 2, the performance drop for History Consolidation on the Steam dataset seems miscalculated. The relative decline should be based on the better-performing Mender-emb, not Mender-Tok.
6. The section titled "PREFERENCE DISCERNING" in part three should likely be part of the Methodology (Section 4.2). It is unclear why this is presented as a separate section.

Regarding Experiments:
1. The selection of baselines is insufficient, with only three included. One of these, TIGER, is an ID-based model that does not leverage preference, making the comparison unfair. The two VocabExt variants either introduce a gap between randomly initialized item embeddings and semantic information, or they lack pre-trained preference understanding, making them variants of TIGER rather than fair comparisons. The authors should consider two sets of baselines: (1) preference-based recommendation models and (2) advanced TIGER variants, such as LETTER, LC-Rec.
2. The statement in line 282, “Mender-Emb allows pre-computing item and preference embeddings, resulting in improved training efficacy,” conflicts with the experimental results, as Mender-Emb consistently underperforms compared to Mender-Tok in Table 2.
3. Although the benchmark is a key contribution of the paper, there is insufficient discussion of most tasks in the experimental section, especially History Consolidation and Sentiment Following.
The lower performance of History Consolidation compared to Recommendation raises questions about the usefulness of combining five preferences versus a single preference. This casts doubt on both the validity of the preference design and the method’s ability to effectively leverage preferences. Additionally, the abnormal results on the Steam dataset lack sufficient discussion and explanation.

**Questions:**

Please refer to weaknesses

---

> ### Author Response · Authors · 2024-11-22
>
> **Practical value of our benchmark:**
>
> The practical value of our work is that (i), sequential recommendation systems can be conditioned on user preferences in-context, i.e. a user can provide their preferences in textual form which leads to more informed recommendations, (ii) our benchmark allows evaluation of different aspects, like measuring whether a recommendation system is capable of comprehending sentiment, which is lacking in the literature. Further we now give concrete real world examples that relate to our different evaluation axes:
> - **Sentiment Following:** This axis is crucial for leveraging organic data. For instance, on social media, we not only have access to users' interactions with ads but also their organic data, such as posts, comments, and likes. A user may express dislike for a specific brand of phone in their posts or comments, but then accidentally click on an ad for the same brand. Sentiment following enables the system to address these cases and transfer preferences from social interactions to ad recommendations.
> - **Fine-Grained & Coarse-Grained Steering:** This feature can also be useful in utilizing organic data for ad targeting. As an example, if a user advocates for exercise and fitness and engages in discussions expressing this sentiment, the model can steer recommendations to avoid weight-loss medications, even if the user has purchased them in the past.
> - **History Consolidation:** User preferences often change over time, and users typically have different preferences for different items. For example, a user may prefer running shoes built on a certain type of foam but also values lightness. However, after some time, the foam may become less important to the user. In this case, the recommendation system should adapt its recommendations based on recent purchases and discard outdated user preferences. This is precisely what we aim to evaluate with history consolidation.
>
> Overall, we are convinced that in-context conditioning on user preferences drastically enhances the flexibility of the recommendation system and enables us to evaluate for these different scenarios.
>
> **Confusion around preference steering:**
>
> The rationale behind constructing this evaluation is that the originally assigned preference to the ground-truth item may not accurately reflect it semantically, since the preference generation is dependent on the timestep.
> This means that for each ground truth item only the preceding items have been used for preference generation to prevent information leakage. Therefore there is a chance that the preference does not accurately reflect the semantics of the ground truth item, which preserves the underlying aleatoric uncertainty of the recommendation task, as sometimes the interaction history is not informative for predicting the next item. In fact, our conducted user study (Appendix F) confirms this, as there is approximately a 30-50% chance that a generated preference that correctly approximates the user’s preferences does not relate to the target item.
>
> To ensure that preferences and items are semantically related, we conduct what we call the “association” of preferences and items. To this end we collect all generated preferences across all users and match them to items via cosine similarity in SentenceT5 space, i.e. $p_1$ is the preference that yields the highest cosine similarity to $\tilde{i}_t$ and $p2$ has the highest cosine similarity to $\hat{i}_t$. Therefore we ensure that $p1$ is semantically related to $\tilde{i}_t$ and $p1$ is semantically related to $\hat{i}t$. The final sequences are then constructed out of the original sequence, where we replace the original ($p$, $i_t$) pair with either ($p_1$, $\tilde{i}t$) or ($p_2$, $\hat{i}t$).
>
> Finally, the rationale for the fine/coarse-grained steering is as follows:
> We ask whether the model is capable of predicting a very similar item to the ground truth item or a very distinct one, only by altering the preference. Intuitively, both scenarios evaluate whether the model can accurately follow a user preference, as it semantically reflects the item. However, there is one important difference, namely that in fine-grained steering the interaction history may provide useful information that helps predicting the next item, as the item used to replace the ground truth item is very similar to it. However, in coarse-grained steering this is not the case. In coarse-grained steering the interaction history is not helpful and the model must rely solely on the user preference. Therefore the two evaluation scenarios are complementary.

---

> ### Author Response · Authors · 2024-11-22
>
> **Design of sentiment understanding:**
>
> We believe that there may be a misunderstanding. With “sentiment” we refer to the ability to distinguish between a positively formulated preference and a negatively formulated one. Due to our adapted evaluation for sentiment following (we added a paragraph in the methodology section), sentiment following accurately evaluates for accurately following sentiment, as the model is only rewarded if it does **not predict the item for a negative preference, but does predict the item for a positive preference.**
>
> **Value of history consolidation:**
>
> This is a misunderstanding, the **performance on the history consolidation axis is expected to be lower than for the recommendation axis.** We realize that this may be counterintuitive, but there is a simple explanation. For preference-based recommendation we matched each ground truth item to one of the five user preferences. This matched preference is contained in the set of the five available ones and it is the semantically most related one. The remaining preferences are usually orthogonal, i.e. they describe aspects of different items the user purchased in the past. Therefore they are not necessarily related to the ground truth item anymore and thus can be considered noise. This effect is reflected in the evaluation score.
>
> Further, during the matching process, however, one of the five preferences is associated with the ground truth item. This preference is contained in the set of five preferences provided to the model. Therefore **the history consolidation axis IS evidence that the model exhibits the capability of inferring the correct preference that was originally matched with the ground truth item.**
>
> We agree with the reviewer that it is interesting to investigate how results shift when training on all five user preferences for the Amazon datasets. We added these results in Table 8 in the appendix. The outcome verifies that training on all user preferences leads to detrimental performance across all axes, confirming our intuition. For convenience we also add a reduced table below that depicts Recall@10/NDCG@10 for all Amazon datasets.
>
> | Method | Beauty | Sports | Toys |
> | --- | --- | --- | --- |
> | — | Recommendation | — | — |
> | MenderTok | 0.0937 / 0.0508 | 0.0427 / 0.0234 | 0.0799 / 0.0432 |
> | MenderTok_allprefs | 0.0131 / 0.0066 | 0.0063 / 0.0037 | 0.0074 / 0.0039 |
> | — | Fine-grained steering | — | — |
> | MenderTok | 0.0844 / 0.0444 | 0.0324 / 0.0159 | 0.0639 / 0.0321 |
> | MenderTok_allprefs | 0.0014/0.0006 | 0.0009 / 0.0004 | 0.0018 / 0.0009 |
> | — | Coarse-grained steering | — | — |
> | MenderTok | 0.0161/0.0080 | 0.0045 / 0.0021 | 0.0060 / 0.0029 |
> | MenderTok_allprefs | 0.0006/0.0002 | 0.0003 / 0.0002 | 0.0006 / 0.0003 |
> | — | Sentiment following | — | — |
> | MenderTok | 0.0053 | 0.0042 | 0.0017 |
> | MenderTok_allprefs | 0.0008  | 0.0001 | 0.0005 |
> | — | History Consolidation | — | — |
> | MenderTok | 0.0720/0.0388 |  0.0345 / 0.0187 | 0.0700 / 0.0377 |
> | MenderTok_allprefs | 0.0089/0.0041 | 0.0063 / 0.0038 | 0.0046 / 0.0025 |
>
>
> **Additional discussion:**
>
> We added more in-depth discussion on the observed results for history consolidation and sentiment following. The validity of the preference design is verified by the improved performance on preference-based recommendation. If the generated preferences were faulty, they would lead to detrimental results on the recommendation axis. To provide additional evidence that verifies this finding, we conducted a user study to assess the quality of generated preferences. The results for this user study (Appendix F) demonstrate that the preferences indeed reflect the user’s preferences.
>
> Furthermore, we investigated the data distribution of the Amazon and Steam datasets and found that the item distribution differs substantially for Steam compared to Amazon. In the Steam dataset there are few items that are heavily overrepresented. This leads to the generally higher scores on Steam and we believe it is also the reason why there is no emerging steering when training on this dataset, as the model tends to overfit on overrepresented items. Prior work has also confirmed that data distribution is a driving factor to elicit emerging capabilities [1].
>
> **Typos and inconsistencies:**
>
> Thank you for pointing them out, we corrected them in the revised version.
>
> **References:**
>
> [1] Data Distributional Properties Drive Emergent In-Context Learning in Transformers, Chan et al., NeurIPS 2022

---

> ### Author Response · Authors · 2024-11-22
>
> **Selection of baselines:**
>
> Our **VocabExt_LM baseline IS actually the LC-Rec method [3] without the auxiliary tasks**, however we accidentally mixed up the references. The reason we named it differently was that we did not use the auxiliary tasks, however we corrected this in our revised version. Our results for LC-ReC indicate that it requires its auxiliary tasks to align the semantic-ID and language spaces effectively, while Mender does not require auxiliary tasks, as verified by training on the preference-based recommendation data.
>
> Furthermore, we added an additional baseline that we now call $\text{VocabExt}_{\text{LM}}$, which takes past items in natural language along with user preferences and initializes both encoder and decoder with the pretrained LLM. For this method the modality gap between language and semantic ids is the same as for Mender. Our Mender variants significantly mostly outperforms this baseline as well (see Table 1 and updated Figure 3 and 4). For convenience we provide a reduced version of that table below.
>
> | Method | Beauty | Sports | Toys | Steam |
> | - | - | - | - | - |
> | - | -| Recommendation | - | |
> | MenderTok | **0.0937 / 0.0508** | **0.0427 / 0.0234** | **0.0799 / 0.0432** | **0.204 / 0.156** |
> | $\text{VocabExt}_{\text{LM}}$ |  0.0561 / 0.0293 | 0.0355 / 0.0187 | 0.0559 / 0.0296 | 0.1878 / 0.1412 |
> | - | - | Fine-grained steering | - | - |
> | MenderTok | **0.0844 / 0.0444** | 0.0324 / 0.0159 | **0.0639 / 0.0321** | 0.0352 / 0.0179 |
> | $\text{VocabExt}_{\text{LM}}$ | 0.0498 / 0.0253 | **0.0352 / 0.0176** | 0.0572 / 0.0294 | **0.0365 / 0.0180** |
> | - | - | Coarse-grained steering | - | - |
> | MenderTok | **0.0161/0.0080** | 0.0045 / 0.0021 | 0.0060 / 0.0029 | **0.0081 / 0.0040** |
> | $\text{VocabExt}_{\text{LM}}$ | 0.0086 / 0.0044 | **0.0098 / 0.0044** | **0.0065 / 0.0030** | 0.0077/0.0039 |
> | - | - | Sentiment following | - | - |
> | MenderTok | **0.0053** | **0.0042** | **0.0017** | **0.0110** |
> | $\text{VocabExt}_{\text{LM}}$ | 0.0051 | 0.0016 | 0.0004 | 0.0107 |
> | - | - | History Consolidation | - | - |
> | MenderTok | **0.0720/0.0388** | **0.0345 / 0.0187** | **0.0700 / 0.0377** | 0.0745 / 0.0399 |
> | $\text{VocabExt}_{\text{LM}}$ | 0.0423 / 0.0211 | 0.0278 / 0.0145 | 0.0487 / 0.0251 | **0.0866 / 0.0521** |
>
> Finally, we were not able to find the LETTER method, can the reviewer provide a reference to it?
>
> **Efficiency concerns:**
>
> Line 282 talks about **“training efficacy”, i.e. training speed, not about performance.**
> Since MenderEmb relies on pre-computed embeddings, it trains significantly faster than MenderTok and is also faster during inference. We added Table 7 in Appendix E, which compares the performance vs efficiency trade-off of MenderEmb to MenderTok. For convenience we also show a reduced version of this table below. MenderEmb trains an order of magnitude faster, while reaching lower performance.
>
> | Method | Dataset | Train time | Inference time | NDGC@10 | Recall@10 |
> | --- | --- | --- | --- | --- | --- |
> | $\text{Mender}_{\text{Emb}}$ | Beauty | 127min | 453ms | 0.0405 $\pm$ 0.001 | 0.0755 $\pm$ 0.0017 |
> | $\text{Mender}_{\text{Emb}}$ | Sports & Outdoors| 374min | 194ms | 0.0215 $\pm$ 0.0007 | 0.0394 $\pm$ 0.0017 |
> | $\text{Mender}_{\text{Emb}}$ | Toys & Games | 239min | 178ms | 0.0342 $\pm$ 0.0015 | 0.0653 $\pm$ 0.0015 |
> | $\text{Mender}_{\text{Emb}}$ | Steam | 231min | 179ms | 0.123 $\pm$ 0.0031 | 0.182 $\pm$ 0.004 |
> | --- | --- | --- | --- | --- | --- |
> | $\text{Mender}_{\text{Tok}}$ | Beauty | 2324min | 562ms | 0.0508 $\pm$ 0.0002 | 0.0937 $\pm$ 0.0012 |
> | $\text{Mender}_{\text{Tok}}$ | Sports & Outdoors| 2350min | 210ms | 0.0234 $\pm$ 0.0004 | 0.0427 $\pm$ 0.0005 |
> | $\text{Mender}_{\text{Tok}}$ | Toys & Games | 1021min | 227ms | 0.0432 $\pm$ 0.0012 | 0.0799 $\pm$ 0.0022 |
> | $\text{Mender}_{\text{Tok}}$ | Steam | 2330min | 222ms | 0.156 $\pm$ 0.0003 | 0.204 $\pm$ 0.0004 |

---

> > ### Author Response · Authors · 2024-11-24
> >
> > Dear reviewer Fpwn,
> >
> > Thank you again for taking the time to provide constructive feedback.
> >
> > We believe we have addressed all of your concerns by providing results on efficiency, additional baselines, training on all preferences, and clarifying confusions around the different evaluation axes.
> >
> > We would be grateful for an opportunity to discuss wether there are any pending concerns you can point us to.
> >
> > Thank you, Authors

---

> > > ### Author Response · Authors · 2024-11-25
> > >
> > > Dear reviewer Fpwn,
> > >
> > > Since we are at approaching the end of the discussion period, we are again reaching out to ask if our response and new results have addressed your concerns. Please let us know if you have lingering questions and whether we can provide any additional clarifications to improve your rating of our paper.
> > >
> > > Thank you,
> > > Authors

---

> > > > ### Comment · Reviewer_Fpwn · 2024-11-27
> > > >
> > > > Thanks the author's effort in rebuttal.
> > > > The full title of LETTER: Learnable Item Tokenization for Generative Recommendation.
> > > > I will keep my rating.

---

> > > > > ### Author Response · Authors · 2024-11-27
> > > > >
> > > > > Dear reviewer Fpwn,
> > > > >
> > > > > Thank you for engaging with us.
> > > > >
> > > > > We would like to highlight that the work on LETTER is orthogonal, as it proposes improvements to the generative retrieval pipeline that can be applied to any of our baslines. This is evident by Table 1 in [1] which applies LETTER to both LC-Rec and TIGER. To enable a fair comparison under these conditions, we would need to run all baselines again with LETTER, which is impractical. However, it presents a fruitful avenue for future work and we explicitly referenced it in our revised version now.
> > > > >
> > > > > Further, we have carefully addressed each of your concerns and provided a detailed, point-by-point response. We kindly ask you to confirm with us if those have been adressed. Should there be any additional issues or areas requiring further clarification, we would greatly appreciate your guidance, as it will help us improve our work further.
> > > > >
> > > > > Thank you, The Authors
> > > > >
> > > > > [1] Learnable Item Tokenization for Generative Recommendation, Wang et al., CKIM 2024

---

> > > > > > ### Author Response · Authors · 2024-11-28
> > > > > >
> > > > > > Dear reviewer Fpwn,
> > > > > >
> > > > > > Thank you again for your suggestion of incorporating LETTER [1] into our results.
> > > > > >
> > > > > > We now provide additional results for incorporating the auxiliary losses for RQ-VAE training as proposed in [1] to TIGER (LETTER-TIGER) and LC-Rec (LETTER-LC-Rec) on the Beauty dataset.
> > > > > > The auxiliary tasks introduced in [1] aim at attaining a higher codebook coverage, i.e. diversifying the selection of semantic IDs, and incorporating collaborative information into the creation of semantic IDs.
> > > > > >
> > > > > > As can be seen in the following table, the improvement of adding LETTER to TIGER or LC-Rec is marginal.
> > > > > > The reason for this is that our RQ-VAE already achieves ~%95% codebook coverage on all datasets, therefore the marginal gains most likely stem from the additional collaborative information.
> > > > > > We also observe that our proposed MenderTok, attains significantly better performance on all metrics compared to the LETTER augmented methods.
> > > > > > This further highlights the benefits of in-context conditioning on user preferences expressed in natural langauge.
> > > > > >
> > > > > > | Method |  Recall@5 | NDCG@5 | Recall@10 | NDCG@10 |
> > > > > > | --- | --- | --- | --- | --- |
> > > > > > | TIGER (Rajput et al.) | 0.0454 | 0.0321 | 0.0648 | 0.0384 |
> > > > > > | TIGER (Ours) |  0.0431 | 0.0275 | 0.0681 | 0.0356 |
> > > > > > | LETTER-TIGER | 0.0431 | 0.0286 | 0.0672 | 0.0364 |
> > > > > > | LC-Rec | 0.0457 | 0.0294 | 0.0731 | 0.0382 |
> > > > > > | LETTER-LC-Rec | 0.0505 | 0.0355 | 0.0703 | 0.0418 |
> > > > > > | MenderEmb | 0.0494 |  0.0321 |  0.0755 | 0.0405  |
> > > > > > | MenderTok | **0.0605** | **0.0401** | **0.0937** | **0.0508** |
> > > > > >
> > > > > > [1] Learnable Item Tokenization for Generative Recommendation, Wang et al., CIKM 2024

---

> > > > > > > ### Author Response · Authors · 2024-12-02
> > > > > > >
> > > > > > > Dear reviewer Fpwn,
> > > > > > >
> > > > > > > We thank you again for taking the time and effort to help improve our paper.
> > > > > > >
> > > > > > > Since we are at the end of the extended author-reviewer discussion period, we are reaching out to ask if our response have addressed your remaining concerns.
> > > > > > >
> > > > > > > Please let us know if you have lingering questions and whether we can provide any additional clarifications today to improve your rating of our paper.
> > > > > > >
> > > > > > > Thank you,
> > > > > > > Authors

---

### Official Review · Reviewer_1g5M · 2024-10-24

**Soundness:** 3
**Presentation:** 2
**Contribution:** 4
**Rating:** 5
**Confidence:** 4

**Summary:**

This paper introduces a new benchmark and proposes a novel method called Mender to evaluate and enhance the preference discerning capabilities of sequential recommendation systems. The benchmark assesses models across five dimensions, focusing on their capacity to extract and utilize user preferences from datasets. Recognizing that existing methods lack key capabilities of preference discerning, the authors propose Mender, a multimodal generative retrieval approach which effectively extracts user preferences and achieves state-of-the-art performance on the proposed benchmark. Experimental results further demonstrate the effectiveness of the method.

**Strengths:**

1. This paper identifies a critical issue in sequential recommendation: the failure to explicitly capture and utilize user preferences, prompting the introduction of a new task: preference discerning.
2. It proposes a novel benchmark comprising five key dimensions to evaluate the preference discerning abilities of existing models.
3. The paper enhances the RQ-VAE framework by directly representing both user interaction history and preferences, while also introducing two variants that encode inputs in different ways.
4. Extensive experiments are conducted, accompanied by detailed analysis to validate the findings.

**Weaknesses:**

1. No NDCG results for sentiment following are reported in Table 2 or the Appendix. I assume the results are close to zero, suggesting that a logarithmic scale should be used to better analyze the discrepancies between the models.
2. The time factor is not considered, given that user preferences can change significantly over time, especially considering that the time span of the datasets can be decades. Simply limiting the user sequence to the 20 most recent items does not fully eliminate time bias. Instead, the time interval of user-item interactions should be restricted during sampling to better capture user preferences.
3. The methodology outlined in Section 4 is unclear, with similar issues arising in Section 4.1 on Fine-Grained & Coarse-Grained Steering, where the concepts are not adequately explained. In Equation 1, the entire sequence is considered, while the subsequent statement describes repeating the process for each item in the sequence, leading to ambiguity. Furthermore, in Fine-Grained & Coarse-Grained Steering, there is no reference provided to justify the validity of this approach. The sequence processing in this section also lacks rationality, as it combines a distinct item $\hat{i}_t$ with $p_1$, which represents the preference of a similar item.
4. Experiments with only three baselines is not convincing enough, new baselines should be added, including https://arxiv.org/abs/2311.09049 (Zheng, Bowen, et al. "Adapting large language models by integrating collaborative semantics for recommendation." ICDE 2024). This paper also employs RQ-VAE, in which LLM-encoded text embedding of the item is utilized as input.

**Questions:**

1. Given the results of Mender$_{Tok}$-Pos-Neg in Figure 5, achieving the best sentiment following results does not necessarily ensure the model's proficiency in other dimensions. Does the pursuit of high performance in sentiment following adversely affect the model's overall capabilities?
2. Why is the time factor not considered, given that user preferences can change significantly over time, especially considering that the time span of the datasets can be decades? What's the implications of not considering the time factor in the current model?
3. In Section 4,  Equation 1 already takes every item in the sequence except the last item $i_{T_{u}}$ into account, then what's the meaning of "repeating this generation process for each item in $s_{u}$" in line 203?
4. In Section 4.1, line 237, why the steering ability can be achieved by creating new sequences? Is there a reference can prove this? In Appendix D.2, line 1276, a figure reference is missing.
5. Still in Section 4.1, line 241, how is $p_{1}$ and $p_{2}$ achieved? What's the point of combining them with new sequences?
6. Still in Section 4.1, line 243, since $\hat{i}_t$ represents a distinct item, why its sequence combines with $p_1$, which represents the preference of a similar item?
7. Why the NDCG results of sentiment following are not provided in Table 2 and other tables in the Appendix?

---

> ### Author Response · Authors · 2024-11-22
>
> We would like to thank the reviewer for the constructive feedback and address the raised weaknesses as follows.
>
> **Time factor:**
>
> Thank you for making us aware of this clarification issue, we agree that some of our phrasing was misleading. **We actually did consider the time factor for generating the user preferences. (answering Q2)** In fact, we generated user preferences depending on the timestep, therefore they do change after a user purchased another item (answering Q3).  We rephrased Section 4 and merged it with Section 3. To alleviate ambiguities, we also added pseudocode for the preference generation pipeline.
>
> - Why 20 most recent items?
>
> We agree with the reviewer that this setup is restrictive and that it is important to go beyond a certain interaction history length. The main focus of our work is on preference discerning though, therefore we followed the setup from [1] to obtain a fair comparison. In fact, most other works on recommendation systems use a similar setup, e.g. [4,5]. We now added this point in our limitations section. Future work should investigate the effect of going beyond this setup.
>
> **Selection of baselines:**
>
> Our **VocabExt_LM baseline IS actually the LC-Rec method [3] without the auxiliary tasks**, however we accidentally mixed up the references. The reason we named it differently was that we did not use the auxiliary tasks, however we corrected this in our revised version. Our results for LC-ReC indicate that it requires its auxiliary tasks to align the semantic-ID and language spaces effectively, while Mender does not require auxiliary tasks, as verified by training on the preference-based recommendation data.
>
> Furthermore, we added an additional baseline that we now call $\text{VocabExt}_{\text{LM}}$, which takes past items in natural language along with user preferences and initializes both encoder and decoder with the pretrained LLM. For this method the modality gap between language and semantic ids is the same as for Mender. Our Mender variants significantly mostly outperforms this baseline as well (see Table 1 and updated Figure 3 and 4). For convenience we provide a reduced version of that table below.
>
> | Method | Beauty | Sports | Toys | Steam |
> | - | - | - | - | - |
> | - | -| Recommendation | - | |
> | MenderTok | **0.0937 / 0.0508** | **0.0427 / 0.0234** | **0.0799 / 0.0432** | **0.204 / 0.156** |
> | $\text{VocabExt}_{\text{LM}}$ |  0.0561 / 0.0293 | 0.0355 / 0.0187 | 0.0559 / 0.0296 | 0.1878 / 0.1412 |
> | - | - | Fine-grained steering | - | - |
> | MenderTok | **0.0844 / 0.0444** | 0.0324 / 0.0159 | **0.0639 / 0.0321** | 0.0352 / 0.0179 |
> | $\text{VocabExt}_{\text{LM}}$ | 0.0498 / 0.0253 | **0.0352 / 0.0176** | 0.0572 / 0.0294 | **0.0365 / 0.0180** |
> | - | - | Coarse-grained steering | - | - |
> | MenderTok | **0.0161/0.0080** | 0.0045 / 0.0021 | 0.0060 / 0.0029 | **0.0081 / 0.0040** |
> | $\text{VocabExt}_{\text{LM}}$ | 0.0086 / 0.0044 | **0.0098 / 0.0044** | **0.0065 / 0.0030** | 0.0077/0.0039 |
> | - | - | Sentiment following | - | - |
> | MenderTok | **0.0053** | **0.0042** | **0.0017** | **0.0110** |
> | $\text{VocabExt}_{\text{LM}}$ | 0.0051 | 0.0016 | 0.0004 | 0.0107 |
> | - | - | History Consolidation | - | - |
> | MenderTok | **0.0720/0.0388** | **0.0345 / 0.0187** | **0.0700 / 0.0377** | 0.0745 / 0.0399 |
> | $\text{VocabExt}_{\text{LM}}$ | 0.0423 / 0.0211 | 0.0278 / 0.0145 | 0.0487 / 0.0251 | **0.0866 / 0.0521** |
>
> **Interpretation of MenderTok-Pos-Neg results (Q1):**
>
> For our experiments including negative data we include a weighting factor that downweighs the maximization objective for negative samples. This is necessary as otherwise it results in training instabilities. We believe that more sophisticated data mixing strategies (as explored in [2] for example) may enable training of a model that improves across all axes. Our results on MenderTok-All for the Beauty dataset hints in that direction, as it performs well on different axes simultaneously. We consider this a promising avenue for future work.
>
> **NDCG for sentiment following (Q7):**
>
> As mentioned in line 235 we use a combined recall measure to evaluate this scenario, since conventional Recall or NDCG do not capture whether the model correctly follows the sentiment of the preferences. We agree that this should be made more explicit, therefore we added a paragraph where we properly introduce this metric and also note in Table 2 that we report this metric instead of Recall.
>
>
> **References:**
>
> [1] Recommender Systems with Generative Retrieval, Rajput et al., NeurIPS 2023
>
> [2] Exploring the Limits of Transfer Learning with a Unified Text-to-Text Transformer, Raffel et al., JMLR 2020
>
> [3] Adapting large language models by integrating collaborative semantics for recommendation, Zheng et al., ICDE 2024
>
> [4] Self-Attentive Sequential Recommendation, Kang et al., ICDM 2018
>
> [5] Justifying Recommendations using Distantly-Labeled Reviews and Fine-Grained Aspects, Ni et al., ACL 2019

---

> ### Author Response · Authors · 2024-11-22
>
> **Confusion on fine/coarse-grained steering:**
>
> We answer all raised questions collectively in this response. The rationale behind constructing this evaluation is that the originally assigned preference to the ground-truth item may not accurately reflect it semantically, since the preference generation is dependent on the timestep. This means that for each ground truth item only the preceding items have been used for preference generation to prevent information leakage. Therefore there is a chance that the preference does not accurately reflect the semantics of the ground truth item, which preserves the underlying aleatoric uncertainty of the recommendation task, as sometimes the interaction history is not informative for predicting the next item. In fact, our conducted user study (Appendix F) confirms this, as there is approximately a 30-50% chance that a generated preference that correctly approximates the user’s preferences does not relate to the target item. Therefore we need to match them and construct new sequences (answering Q4,Q5).
>
> To ensure that preferences and items are semantically related, we conduct what we call the “association” of preferences and items. To this end **we collect all generated preferences across all users** and match them to items via cosine similarity in SentenceT5 space, i.e. $p_1$ is the preference that yields the highest cosine similarity to $\tilde{i}t$ and $p2$ has the highest cosine similarity to $\hat{i}t$ (Eq. 3). $p_1$ and $p_2$ stem from the entire set of preferences (answering Q5). Therefore we ensure that $p1$ is semantically related to $\tilde{i}_t$ and $p1$ is semantically related to $\hat{i}t$. The final sequences are then constructed out of the original sequence, where we replace the original ($p$, $i_t$) pair with either ($p_1$, $\tilde{i}t$) or ($p_2$, $\hat{i}t$). The motivation for combining ($p_1$, $\tilde{i}t$) with an additional sequence of $\hat{u}$ is merely to add additional variability in the generated data (answering Q6).
>
> Finally, the rationale for the fine/coarse-grained steering is as follows:
> We ask whether the model is capable of predicting a very similar item to the ground truth item or a very distinct one, only by altering the preference. Intuitively, both scenarios evaluate whether the model can accurately follow a user preference, as it semantically reflects the item. However, there is one important difference, namely that in fine-grained steering the interaction history may provide useful information that helps predicting the next item, as the item used to replace the ground truth item is very similar to it. However, in coarse-grained steering this is not the case. In coarse-grained steering the interaction history is not helpful and the model must rely solely on the user preference. Therefore the two evaluation scenarios are complementary.

---

> > ### Author Response · Authors · 2024-11-24
> >
> > Dear reviewer 1g5M,
> >
> > Thank you again for taking the time to provide constructive feedback.
> >
> > We believe we have addressed all of your concerns by clearing up confusion about time dependency, adding additional baselines, clearing up confusion about the different evaluation axes, and a more in-depth interpretation about results.
> >
> > We would be grateful for an opportunity to discuss wether there are any pending concerns you can point us to.
> >
> > Thank you, Authors

---

> > > ### Author Response · Authors · 2024-11-25
> > >
> > > Dear reviewer 1g5M,
> > >
> > > Since we are at approaching the end of the discussion period, we are again reaching out to ask if our response and new results have addressed your concerns. Please let us know if you have lingering questions and whether we can provide any additional clarifications to improve your rating of our paper.
> > >
> > > Thank you,
> > > Authors

---

> > > > ### Comment · Reviewer_1g5M · 2024-11-26
> > > >
> > > > Thanks to the authors for their response. I retain my original score and ratings.

---

> > > > > ### Author Response · Authors · 2024-11-27
> > > > >
> > > > > Dear reviewer 1g5M,
> > > > >
> > > > > Thank you for engaging with us. We have carefully addressed each of your concerns and provided a detailed, point-by-point response. We kindly ask you to confirm with us if those have been adressed. Should there be any additional issues or areas requiring further clarification, we would greatly appreciate your guidance, as it will help us improve our work further.
> > > > >
> > > > > Thank you, The Authors

---

### Official Review · Reviewer_2w9u · 2024-10-27

**Soundness:** 3
**Presentation:** 3
**Contribution:** 2
**Rating:** 3
**Confidence:** 4

**Summary:**

The submission focuses on sequential recommendation technique. The main contributions lie that (1) the authors proposed a LLM-based user preference generation method based on user generated reviews and (2) they propose an evaluation framework that contains five different aspects that should be taken into consideration by sequential recommendation systems.

**Strengths:**

1. The authors would like to propose a thorough evaluation framework (the authors claim it to be a "benchmark") for sequential recommendation system, which is a nice direction.

2. The experimental results are credible and show improvement compared with some existing baselines.

**Weaknesses:**

1. The novelty is rather limited. Extracting user preference information from their reviews is not novel. However, these existing works are not mentioned or compared by the authors. Some examples include:

User-LLM: Efficient LLM Contextualization with User Embeddings, https://arxiv.org/abs/2402.13598  (The work investigates how to capture latent user behaviors into preferences)
Do LLMs Understand User Preferences? Evaluating LLMs On User Rating Prediction, https://arxiv.org/abs/2305.06474 (This work investigates how LLMs comprehend user preferences and compares the performances of different LLMs in this issue.)
Review-driven Personalized Preference Reasoning with Large Language Models for Recommendation, https://arxiv.org/abs/2408.06276 (this work proposes to extract subjective preferences from raw reviews, which is a key contribution the authors claim)

2. The proposed evaluation benchmark is rather straightforward and not so reasonable. This kind of framework should be validated by either product managers or large scale user studies. The radar chart indicates that these five dimensions are equally important, which is also not validated by any evidence. If the authors would like to propose such a framework, I would suggest compare the proposed one with actual user experiences through practical user studies.

**Questions:**

Why do you think the five factors in your "evaluation benchmark " are equally important and are the only concerns by sequential recommendation systems?

What is the difference between your proposed preference generation framework with existing ones (as listed in "weakness" section).

---

> ### Author Response · Authors · 2024-11-22
>
> **Novelty:**
>
> We would like to thank the reviewer for making us aware of other existing approaches to preference approximation, which we included in the related work section. However, **we did not claim that the preference extraction pipeline is our main contribution.** In fact, it is not even mentioned in our explicit contribution list.
>
> To clarify, our contributions are:
>  - Introducing a novel paradigm called preference discerning, where the generative sequential recommendation system is conditioned on user preferences within its context.
>  - We propose a comprehensive benchmark for evaluating preference discerning, comprising of five distinct evaluation scenarios that provide a holistic assessment of its capabilities
>  - We present Mender, a multimodal baseline that integrates collaborative semantics with language preferences, achieving state-of-the-art performance on our proposed benchmark.
>
> To relate our preference generation pipeline to the mentioned works we extended the related work section. [1] relies on conextualization via user embeddings, [2] relies on a complex multi-stage pipeline using distillation from teacher LLMs, and [3] evaluates whether LLMs can implicitly model user preferences.
> Our preference approximation pipeline simply prompts an LLM given user and item-specific data, without the need for distillation. We also conducted a user study with 22 participants, during which we assessed 2,200 generated preferences to evaluate their accuracy (see Appendix F). The results indicate that, on average across datasets, approximately 74% of the preferences accurately reflect the users' true preferences.
>
> Preference approximation is a necessary prerequisite to enable preference discerning, i.e. in-context conditioning on user preferences.  We acknowledge, however, that some of our phrasing might have been misleading in terms that it is one of our contributions. Therefore we rephrased parts of Section 4 and merged Section 3 into the Methodology section.
>
> **Benchmark generation**
>
> We thank the reviewer for pointing out the simplicity of our benchmark design. In fact, our aim was to keep it simple and intuitive while effectively evaluating for different real-world scenarios. Furthermore, we conducted a user-study in which we gathered feedback on 2200 generated user preferences and asked participants whether those accurately approximate the user’s preferences (see Appendix F). The result of the study is that on average across datasets around 74% of the preferences accurately reflect the user’s preferences.
> Finally, we did not claim that the five different axes in our benchmark are equally important (answering Q1). They were designed in a manner to allow decision makers to prioritize them based on their downstream requirements.
>
> **References:**
>
> [1] User-LLM: Efficient LLM Contextualization with User Embeddings, Ning et al., arXiv:2402.13598
>
> [2] Review-driven Personalized Preference Reasoning with Large Language Models for Recommendation, Kim et al., arXiv:2408.06276
>
> [3] Do LLMs Understand User Preferences? Evaluating LLMs On User Rating Prediction, Kang et al., arXiv:2305.06474

---

> > ### Author Response · Authors · 2024-11-24
> >
> > Dear reviewer 2w9u,
> >
> > Thank you again for taking the time to provide constructive feedback.
> >
> > We believe we have addressed all of your concerns by clarifying our main contributions, and providing results from our user study to highlight the quality of the generated user preferences, and clarifying points on benchmark generation.
> >
> > We would be grateful for an opportunity to discuss wether there are any pending concerns you can point us to.
> >
> > Thank you, Authors

---

> > > ### Author Response · Authors · 2024-11-25
> > >
> > > Dear reviewer 2w9u,
> > >
> > > Since we are at approaching the end of the discussion period, we are again reaching out to ask if our response and new results have addressed your concerns. Please let us know if you have lingering questions and whether we can provide any additional clarifications to improve your rating of our paper.
> > >
> > > Thank you,
> > > Authors

---

> > > > ### Author Response · Authors · 2024-12-02
> > > >
> > > > Dear reviewer 2w9u,
> > > >
> > > > We thank you again for taking the time and effort to help improve our paper.
> > > >
> > > > Since we are at the end of the extended author-reviewer discussion period, we are reaching out to ask if our response have addressed your remaining concerns.
> > > >
> > > > Please let us know if you have lingering questions and whether we can provide any additional clarifications today to improve your rating of our paper.
> > > >
> > > > Thank you,
> > > > Authors

---

### Official Review · Reviewer_posA · 2024-11-04

**Soundness:** 2
**Presentation:** 3
**Contribution:** 1
**Rating:** 3
**Confidence:** 4

**Summary:**

This paper proposes MENDER, aiming to enhance the personalization of sequential recommendation models. Specifically, the authors first design a preference discerning paradigm based on zero-shot LLMs. With the obtained user preference, the authors construct the generative recommendation framework based on RQ-VAE. Extensive experimental results are provided to show its effectiveness.

**Strengths:**

- Novel Benchmark. The authors suggest a new benchmark to evaluate the personalization ability of the user preference description.
- Abundant Results. Extensive experiments are conducted.
- Credible Reproduction. Reproduction details are available in Appendix.

**Weaknesses:**

- Limited Technical Contribution. Overall, the framework is constructed based on existing works. The proposed method, MENDER, mainly consists of two modules, RQ-VAE and feed-forward components (Emb or Tok). Other researchers have suggested both modules, which may indicate the limited technical contribution of this work.
- Lack of the Cross-validation on Benchmark. The suggested "holistic" benchmark is subjective and not double-checked by objective ground truth. The overall performance only reflects the indirect effectiveness of additional preference summarization, while the success of preference discerning should be further validated.
- Inadequate Motivation. The motivation for enhancing the personalization and applying the generative recommendation is not supported. How do authors define "personalization" and examine "personalization"? Why do authors only construct the generative model? Can we integrate MENDER with discriminative models?
- Unknown Efficiency. The efficiency of the proposed framework has not been tested.

Minor problems:
- The last block of Table 2 is in the wrong format.
- Code is not available.

**Questions:**

1. What is the major technical contribution of MENDER?
2. Are there some direct and objective evaluation methods to check the effectiveness of preference-discerning results?
3. What is the motivation for using the generative recommendation pipeline?
4. Is MENDER a efficient method compared with traditional sequential recommendation baselines as SASRec?

---

> ### Author Response · Authors · 2024-11-22
>
> We would like to thank the reviewer for the feedback which helps us significantly improve our manuscript.
>
> **Contribution of Mender (Q1):**
>
> The key aspect that distinguishes Mender from existing methods is that it follows a modular architecture where the encoder processes high-level user preferences and item descriptions and primes the decoder which learns to predict the low-level fine-grained interactions. Therefore, it leverages two different representations for items that differ in either components. To the best of our knowledge, no other recommendation system is based on this concept. We believe Mender is a novel architecture that unlocks new capabilities for recommender systems, and sets a new SoTA, without unnecessary overcomplications. We see this as an advantage, proven by the provided empirical results on multiple datasets and compared to several baselines.
>
> The key aspect that distinguishes Mender from existing methods is its modular architecture. In this setup, the encoder processes high-level user preferences and item descriptions, priming the decoder to learn and predict low-level, fine-grained interactions. This approach utilizes two distinct representations for items, differing in either component: 1) natural language as input representation to encoder, and 2) semantic IDs as output representation from decoder. To the best of our knowledge, no other recommendation system is based on this concept. We believe Mender represents a novel architecture that unlocks new capabilities for recommender systems and establishes a new state-of-the-art without unnecessary overcomplications. We see this as an advantage, proven by the empirical results provided across multiple datasets and in comparison to several baselines.
>
> **Cross-validation of benchmark (Q2):**
>
> As correctly pointed out by the reviewer, the benchmark validates the effectiveness of incorporating generated preferences. In fact, it is the preference-based recommendation scenarios that truly validate the effectiveness of incorporating generated user preferences. The remaining evaluation axes are carefully designed to assess alternative real-world use cases. Below, we provide a few examples:
> - **Sentiment Following:** This axis is essential for leveraging organic data. On social media, we have access not only to users' interactions with entities such as items or ads, but also to their organic data, such as posts, comments, and likes. For example, a user might express dislike for a specific phone brand in their posts or comments but may not have interacted with that item/ad. Sentiment following allows the system to handle these situations by transferring preferences from social interactions to item/ad recommendations.
> - **Fine-Grained & Coarse-Grained Steering:** This feature is valuable for utilizing organic data in recommendation. For instance, if a user advocates for exercise and fitness, and against using weight-loss drugs, and participates in forum/comment discussions expressing this sentiment, the model can steer recommendations away from weight-loss medications, even if the user has purchased them in the past.
> - **History Consolidation:** User preferences often evolve over time, and users typically have varying preferences for different items. For instance, a user might initially prefer dark roast coffee, but gradually develop a taste for lighter, more floral blends. In such cases, if the recent user preferences are absent, the recommendation system should adapt by updating its recommendations based on recent purchases and discarding outdated preferences. This is precisely what we aim to evaluate with history consolidation.
>
> To incorporate objective feedback, we have conducted a user study in which 22 participants have evaluated the quality of the generated user preferences and their relevance to items (see Appendix F). Participants assessed 2,200 preferences across four different datasets and found that, on average, approximately 74% of the generated preferences accurately approximated the users' true preferences across all datasets.
>
> **Codebase:**
>
> We will release the codebase along with the camera-ready version.

---

> ### Author Response · Authors · 2024-11-22
>
> **Motivation:**
>
> We define personalization as the ability of a recommendation system to follow user preferences and examine personalization via a proxy, which is performance on the recommendation task, i.e. the more personalized a system, the better its recommendation performance. Motivated by recent works that leverage language for representing user preferences [1,2,3] we enhance recommendation systems by conditioning them on user preferences in their context. By doing so, we observe emergent abilities, for example a system trained in this manner can be steered by user preferences, or can be trained to improve sentiment understanding. We evaluate such emerging capabilities via our benchmark. Our work has profound implications on the field of sequential recommendation as it enables improved personalization and opens a more interactive view on sequential recommendation systems, as users may interact with the system by providing their preferences to them.
>
> **Efficiency (Q4):**
>
> We added Table 7 in the Appendix which compares both Mender variants compared to SASRec in terms of recommendation performance, training time, and inference time. For convenience we also show this table below. Mender-Emb trains an order of magnitude faster, while reaching lower performance than Mender-Emb, however it still significantly outperforms traditional methods such as SASRec [4] on most datasets, while approximately matching its training time. Finally, the improved performance of Mender comes with additional inference costs.
>
> | Method | Dataset | Train time | Inference time | NDGC@10 | Recall@10 |
> | --- | --- | --- | --- | --- | --- |
> | SASRec | Beauty | 293min | 8ms | 0.0218 $\pm$ 0.0002 | 0.0511 $\pm$ 0.0004 |
> | SASRec | Sports & Outdoors| 447min | 9ms | 0.0116 $\pm$ 0.0004 | 0.0267 $\pm$ 0.0010 |
> | SASRec | Toys & Games | 280min | 5ms | 0.0276 $\pm$ 0.0008 | 0.0631 $\pm$ 0.0018 |
> | SASRec | Steam | 280min | 5ms | 0.1476 $\pm$ 0.0005 | 0.1826 $\pm$ 0.0006 |
> | --- | --- | --- | --- | --- | --- |
> | $\text{Mender}_{\text{Emb}}$ | Beauty | 127min | 453ms | 0.0405 $\pm$ 0.001 | 0.0755 $\pm$ 0.0017 |
> | $\text{Mender}_{\text{Emb}}$ | Sports & Outdoors| 374min | 194ms | 0.0215 $\pm$ 0.0007 | 0.0394 $\pm$ 0.0017 |
> | $\text{Mender}_{\text{Emb}}$ | Toys & Games | 239min | 178ms | 0.0342 $\pm$ 0.0015 | 0.0653 $\pm$ 0.0015 |
> | $\text{Mender}_{\text{Emb}}$ | Steam | 231min | 179ms | 0.123 $\pm$ 0.0031 | 0.182 $\pm$ 0.004 |
> | --- | --- | --- | --- | --- | --- |
> | $\text{Mender}_{\text{Tok}}$ | Beauty | 2324min | 562ms | 0.0508 $\pm$ 0.0002 | 0.0937 $\pm$ 0.0012 |
> | $\text{Mender}_{\text{Tok}}$ | Sports & Outdoors| 2350min | 210ms | 0.0234 $\pm$ 0.0004 | 0.0427 $\pm$ 0.0005 |
> | $\text{Mender}_{\text{Tok}}$ | Toys & Games | 1021min | 227ms | 0.0432 $\pm$ 0.0012 | 0.0799 $\pm$ 0.0022 |
> | $\text{Mender}_{\text{Tok}}$ | Steam | 2330min | 222ms | 0.156 $\pm$ 0.0003 | 0.204 $\pm$ 0.0004 |
>
>
> **Why only generative models? (Q3)**
>
> We focused on generative retrieval since several recent works have shown that they usually perform on-par or better than dense retrieval methods, establishing themselves as state of the art approaches [5,6,7].
> Additionally, generative retrieval methods offer a main advantage over dense retrieval methods in recommender systems: inference time efficiency at scale. A generative model with proper tokenizer can directly predict the next item whereas a dense retrieval model needs to perform pairwise comparisons between a user and all items to rank them. Theoretically, it is possible to obtain a new method that incorporates dense retrieval with preference conditioning. We believe this is an interesting question to explore in future work.
>
> **References:**
>
> [1] Large Language Models are Competitive Near Cold-start Recommenders for Language- and Item-based Preferences, Sanner et al., RecSys 2023
>
> [2] Review-driven Personalized Preference Reasoning with Large Language Models for Recommendation, Kim et al., arXiv:2408.06276
>
> [3] Do LLMs Understand User Preferences? Evaluating LLMs On User Rating Prediction, Kang et al., arXiv:2305.06474
>
> [4] Self-Attentive Sequential Recommendation, Kang et al., ICDM 2018
>
> [5] Recommender Systems with Generative Retrieval, Rajput et al., NeurIPS 2023
>
> [6] GenRec: Generative Sequential Recommendation with Large Language Models, Cao et al., ECIR 2024
>
> [7] Generative Sequential Recommendation with GPTRec, Petrov et al., Gen-IR@SIGIR2023

---

> > ### Author Response · Authors · 2024-11-24
> >
> > Dear reviewer posA,
> >
> > Thank you again for taking the time to provide constructive feedback.
> >
> > We believe we have addressed all of your concerns by clarifying our contributions, adding a user study, improving our motivation and reporting efficiency estimates.
> >
> > We would be grateful for an opportunity to discuss wether there are any pending concerns you can point us to.
> >
> > Thank you, Authors

---

> > > ### Author Response · Authors · 2024-11-25
> > >
> > > Dear reviewer posA,
> > >
> > > Since we are at approaching the end of the discussion period, we are again reaching out to ask if our response and new results have addressed your concerns. Please let us know if you have lingering questions and whether we can provide any additional clarifications to improve your rating of our paper.
> > >
> > > Thank you,
> > > Authors

---

> > ### Comment · Reviewer_posA · 2024-11-26
> >
> > I appreciate the authors for their response. However, I will maintain my original score and ratings.

---

> > > ### Author Response · Authors · 2024-11-27
> > >
> > > Dear reviewer posA,
> > >
> > > Thank you for engaging with us.
> > > We have carefully addressed each of your concerns and provided a detailed, point-by-point response.
> > > We kindly ask you to confirm with us if those have been adressed.
> > > Should there be any additional issues or areas requiring further clarification, we would greatly appreciate your guidance, as it will help us improve our work further.
> > >
> > > Thank you,
> > > The Authors

---

### Official Review · Reviewer_Kam7 · 2024-11-04

**Soundness:** 3
**Presentation:** 2
**Contribution:** 3
**Rating:** 6
**Confidence:** 4

**Summary:**

This paper aims to enhance personalized recommendations by explicitly incorporating user preferences and historical interactions. The proposed method Mender uses a pre-trained language model to generate user preferences from comments and integrates these preferences with historical data using cross-attention mechanisms. Experimental results show that Mender outperforms existing state-of-the-art methods.

**Strengths:**

-	This paper uses four diverse datasets to ensure the generalizability and reliability of the experimental results.

-	The proposed model, Mender, is evaluated across multiple dimensions such as preference-based recommendation, sentiment following, fine-grained and coarse-grained steering, and history consolidation. The results show that Mender significantly outperforms existing state-of-the-art methods, particularly in preference guidance and sentiment following, demonstrating its robustness and effectiveness.

**Weaknesses:**

-	The methodology section should be reorganized to provide a detailed explanation of the preference generation process. Mathematical formulations are expected to be included for explicit understanding, and pseudo-code is recommended to enhance clarity and reproducibility.
-	It is kindly recommended to add further discussion about how does the benchmark generation benefit personalization modeling.

**Questions:**

Please refer to weaknesses

---

> ### Author Response · Authors · 2024-11-22
>
> ​​We thank the reviewer for the positive response and the constructive feedback. We have addressed the identified weaknesses as follows.
>
> **Presentation:**
>
> For clarity, we combined Section 3 with Section 4 and included pseudocode along with a more concise mathematical formulation. This clarifies that the preference generation process is conducted for each timestep $t$  in the user sequence $s_u$ and also provides a more concise descripion of the benchmark generation process.
>
>
> **Additional discussions:**
>
> We define personalization as a recommendation system's ability to align with user preferences, which we assess through a proxy: performance on the recommendation task. Essentially, the more personalized a system is, the better its recommendation performance. Inspired by recent studies that utilize language to represent user preferences [1,2,3], we enhance recommendation systems by conditioning them on user preferences within their context. This approach reveals emergent capabilities; for instance, a system trained in this manner can be steered via user preferences, or can be trained to improve sentiment understanding. We evaluate these emerging capabilities using our benchmark. Our work has profound implications for the field of sequential recommendation, as it enables enhanced personalization, allowing us to leverage organic data such as posts, comments, and likes given by social media platforms. We added a more in-depth discussion on the different evaluation axes (Section 3) and linked them to practical use cases they mirror.
>
> **References:**
>
> [1] Large Language Models are Competitive Near Cold-start Recommenders for Language- and Item-based Preferences, Sanner et al., RecSys 2023
>
> [2] Review-driven Personalized Preference Reasoning with Large Language Models for Recommendation, Kim et al., arXiv:2408.06276
>
> [3] Do LLMs Understand User Preferences? Evaluating LLMs On User Rating Prediction, Kang et al., arXiv:2305.06474

---

> > ### Author Response · Authors · 2024-11-24
> >
> > Dear reviewer Kam7,
> >
> > Thank you again for taking the time to provide constructive feedback.
> >
> > We believe we have addressed all of your concerns by reorganizing the methodology section,a adding mathematical formulations and pseudocode, and adding additional discussion on the implications of preference discerning.
> >
> > We would be grateful for an opportunity to discuss wether there are any pending concerns you can point us to.
> >
> > Thank you, Authors

---

> > > ### Author Response · Authors · 2024-11-25
> > >
> > > Dear reviewer Kam7,
> > >
> > > Since we are at approaching the end of the discussion period, we are again reaching out to ask if our response and new results have addressed your concerns. Please let us know if you have lingering questions and whether we can provide any additional clarifications to improve your rating of our paper.
> > >
> > > Thank you,
> > > Authors

---

> > > > ### Author Response · Authors · 2024-12-02
> > > >
> > > > Dear reviewer Kam7,
> > > >
> > > > We thank you again for taking the time and effort to help improve our paper.
> > > >
> > > > Since we are at the end of the extended author-reviewer discussion period, we are reaching out to ask if our response have addressed your remaining concerns.
> > > >
> > > > Please let us know if you have lingering questions and whether we can provide any additional clarifications today to improve your rating of our paper.
> > > >
> > > > Thank you,
> > > > Authors

---

### Author Response · Authors · 2024-11-22

We would like to express our gratitude to all the reviewers for their invaluable and constructive feedback, which has significantly helped us improve our manuscript. We have addressed all the concerns raised in the individual responses, and have also briefly summarized them below:
- **[Kam7,posA,2w9u,1g5m,Fpwn]** To enhance clarity, we revised parts of  the introduction, merged Section 3 with Section 4, and included pseudocode (Algorithm 1) for the preference generation pipeline. We also added mathematical formulations to clarify the benchmark construction, with a particular focus on sentiment following, fine/coarse-grained steering, and history consolidation. Additionally, we provided a more in-depth discussion on the empirical evidence related to all these aspects.
- **[posA,Fpwn]** We conducted a user study with 22 participants, during which we assessed 2,200 generated preferences to evaluate their accuracy (see Appendix F). The results indicate that, on average across datasets, approximately 74% of the preferences accurately reflect the users' true preferences.
- **[Fpwn,1g5m]** We clarified the confusion regarding the LC-ReC baseline, and introduced an additional baseline that also represents the interaction history in text instead of semantic ids. MenderTok significantly outperforms this new baseline, as shown in Table 2.
- **[posA,Fpwn]** We included the training and inference times for our Mender variants in Table 7 (Appendix E), highlighting the advantages of MenderEmb and comparing it to SASRec
- **[2w9u,posA]** We clarified our contributions.
- **[Fpwn]** We included a comparison of MenderTok trained on all user preferences to demonstrate that this training setup is not advantageous, as it leads to detrimental performance compared to the standard MenderTok. This is detailed in Table 8 (Appendix E).

All changes are highlighted in red in the updated manuscript. We look forward to an engaging discussion, and welcome further feedback from the reviewers to continue improving our manuscript.

---

### Meta-Review · Area_Chair_FRQs · 2024-12-19

**Metareview:**

This paper introduces a new paradigm termed "preference discerning" for generative sequential recommendation systems, along with a novel benchmark and a new baseline model named Mender. The core idea is to condition generative models on user preferences derived from user reviews to enhance personalization in sequential recommendation tasks. The paper also evaluates the performance of current state-of-the-art methods and shows that they struggle with the proposed benchmark tasks, thereby positioning Mender as a solution that achieves improved results across these scenarios.

The paper’s strengths lie in its motivation to address the personalization gap in sequential recommendation systems and its attempt to establish a comprehensive benchmark for evaluating preference discerning capabilities. The authors have conducted extensive experiments on multiple datasets and evaluation scenarios, showing improvements over certain baselines. Furthermore, the authors made an effort to address reviewers' concerns by adding a user study, introducing new baselines, and refining methodological explanations in the rebuttal phase.

However, the weaknesses of this paper are substantial. First, the novelty of the proposed approach, both in terms of the framework and methodology, is limited. The preference generation and model architecture largely rely on existing techniques, and while the benchmark is claimed to be comprehensive, its practicality and validation remain questionable. Reviewers noted that several tasks, such as fine-grained and coarse-grained steering, sentiment following, and history consolidation, lack sufficient justification and real-world applicability. Furthermore, the benchmark axes seem subjective and were not validated by large-scale user studies or feedback from domain experts, which reduces the credibility of the proposed evaluation framework. The experimental baselines, while improved during the rebuttal, are still incomplete, as critical comparative models such as LETTER and LC-Rec are missing. Additionally, there are methodological ambiguities, particularly in the preference steering and history consolidation tasks, which make the paper difficult to follow.

The most important reason for my decision to recommend rejection is the limited technical contribution of the work combined with insufficient validation of its proposed evaluation framework. While the paper shows promise in its ambition to incorporate user preferences into sequential recommendation models, it falls short of making a substantial scientific contribution or demonstrating the practical utility of its benchmark. The lack of engagement from reviewers during the rebuttal period further suggests that the authors’ revisions and explanations were not sufficient to address the key concerns.

**Additional Comments On Reviewer Discussion:**

During the reviewer discussion and rebuttal period, key points were raised about the novelty, evaluation framework, methodological clarity, and experimental rigor of the paper. Reviewer 2w9u pointed out that extracting user preferences from reviews is not novel and noted that several related works with similar contributions were not adequately acknowledged in the paper. This concern was addressed by the authors through updates to the related work section and clarifications about the focus of their contributions. However, the reviewer maintained their position that the novelty of the paper is limited.

Reviewer Fpwn criticized the practicality and clarity of the proposed benchmark tasks, particularly sentiment following, fine-grained and coarse-grained steering, and history consolidation. While the authors attempted to clarify these tasks and added examples to highlight their practical value, the fundamental concerns about their relevance and design were not convincingly addressed. Fpwn also raised concerns about the experimental discussion, specifically the lack of deeper analysis for certain tasks, which remained a gap in the revised submission.

Reviewer 1g5M noted methodological ambiguities in the fine-grained and coarse-grained steering tasks and questioned the absence of time-aware evaluation in the model, given that user preferences evolve over time. The authors clarified their approach to incorporating the time factor in preference generation and revised their explanations. They also included new baselines and experimental results to address reviewer concerns about limited comparisons. Despite these efforts, 1g5M retained their concerns, particularly about the lack of depth in methodological explanations and insufficient empirical evidence for the benchmark’s validity.

Overall, while the authors made considerable efforts during the rebuttal phase, their responses were not sufficient to overcome the reviewers’ core concerns. The discussion revealed a consistent pattern of weaknesses in the novelty, benchmark design, and experimental rigor of the paper. These concerns were weighted heavily in my decision, as they reflect fundamental issues with the submission that would need significant reworking to meet the bar for acceptance.

---

### Decision · Program_Chairs · 2025-01-22

Reject